# Hydrothermal alteration of andesitic lava domes can lead to explosive volcanic behaviour

Michael J. Heap [1]*, Valentin R. Troll [2,3], Alexandra R.L. Kushnir[1], H. Albert Gilg [4], Amy S.D. Collinson[5], Frances M. Deegan[2], Herlan Darmawan[6,7], Nadhirah Seraphine[2], Juergen Neuberg [5] & Thomas R. Walter [6]

Dome-forming volcanoes are among the most hazardous volcanoes on Earth. Magmatic outgassing can be hindered if the permeability of a lava dome is reduced, promoting pore pressure augmentation and explosive behaviour. Laboratory data show that acid-sulphate alteration, common to volcanoes worldwide, can reduce the permeability on the sample lengthscale by up to four orders of magnitude and is the result of pore- and microfracture-filling mineral precipitation. Calculations using these data demonstrate that intense alteration can reduce the equivalent permeability of a dome by two orders of magnitude, which we show using numerical modelling to be sufficient to increase pore pressure. The fragmentation criterion shows that the predicted pore pressure increase is capable of fragmenting the majority of dome-forming materials, thus promoting explosive volcanism. It is crucial that hydrothermal alteration, which develops over months to years, is monitored at dome-forming volcanoes and is incorporated into real-time hazard assessments.

[1] Institut de Physique de Globe de Strasbourg (UMR 7516 CNRS, Université de Strasbourg/EOST), 5 rue René Descartes, 67084 Strasbourg, cedex, France. [2] Department of Earth Sciences, Section for Mineralogy, Petrology and Tectonics (MPT), Uppsala University, Uppsala, Sweden. [3] Faculty of Geological Engineering, Universitas Padjajaran (UNPAD), Bandung, Indonesia. [4] Chair of Engineering Geology, Technical University of Munich, 80333 Munich, Germany. [5] School of Earth & Environment, The University of Leeds, Leeds, United Kingdom. [6] GFZ German Research Center for Geosciences, Telegrafenberg, 14473 Potsdam, Germany. [7] Laboratory of Geophysics, Universitas Gadjah Mada, Yogyakarta, Indonesia. *email: heap@unistra.fr

The permeability of a volcanic system exerts a fundamental control on the ability of conduit-filling magma to outgas[1,2]. If magmatic volatiles cannot escape, the pressure inside pores within the magma increases, which is thought to promote explosive volcanic behaviour[3–8]. Lava domes, mounds of blocky lava that form as high-viscosity magma slowly extrudes from the top of a magma-filled conduit[9,10], are intrinsically linked with both magmatic and volatile-driven explosive volcanic activity[11,12]. For example, the growth of a lava dome may act to inhibit outgassing and promote explosive volcanism by: closing shallow-depth outgassing fractures[13,14], diverging slip-lines[15], or plugging the conduit[1,16,17], as seen at Galeras volcano (Columbia) where the emplacement of a lava dome in 1991 led to a decrease in SO2 flux followed by a dome-destroying explosion in 1992[18].

Dome-forming materials are commonly hydrothermally altered by circulating high-temperature fluids[19–21]. Pore- and fracture-filling hydrothermal alteration of a lava dome is considered to lower its permeability, reduce outgassing efficiency, and encourage explosive behaviour[22–25]. For example, recent gas monitoring ($SO_2/CO_2$ and $SO_2$ fluxes) at Poás volcano (Costa Rica) led to models suggesting that hydrothermal sealing may have been the cause of the explosive phreatomagmatic eruption in 2017[26]. Despite the potential importance of lava dome permeability in regulating volcanic outgassing[1,2], and the near-ubiquity of widespread alteration at lava domes[19–21], no studies have thus far provided values for the equivalent permeability of hydrothermally altered lava domes to quantitatively inform volcanic hazard assessments.

It is understood that volcanic character, effusive vs. explosive, depends on many interconnected parameters[8]. Magma flow rate, for example, will dictate the time available for outgassing, cooling, and crystallisation that, in turn, influence magma viscosity[8] and the resultant dome morphology, including the number density and morphology of fractures within the dome[27]. The goal of this contribution is to quantitatively assess whether hydrothermal alteration alone is sufficient to promote explosive volcanic behaviour. Our study shows that hydrothermal alteration can decrease the permeability of a laboratory sample by up to four orders of magnitude. Microstructural observations show that these reductions are a result of pore- and microfracture-filling precipitation of alteration minerals, particularly alunite. We upscale these laboratory measurements to the scale of a lava dome using an effective medium approach and then, using a numerical model, we show that decreases to the equivalent permeability of a dome due to hydrothermal alteration results in an increase in pore pressure within and beneath the dome. Finally, the fragmentation criterion highlights that the predicted increase in pore pressure is capable of fragmenting the majority of dome-forming materials. We conclude that hydrothermal alteration alone can prompt erratic explosive behaviour and, as a result, we recommend that hydrothermal alteration is monitored at active dome-forming volcanoes using geophysical techniques (e.g. electrical and muon tomography) and continuous gas monitoring and is incorporated into real-time hazard assessments at active volcanoes worldwide.

## Results

**Sample collection and description.** The materials for this study were collected from the summit of Merapi volcano, one of the most active and hazardous (>1000 fatalities in the last 150 years) basaltic-andesitic stratovolcanoes in Central Java, Indonesia[28–30]. A new lava dome has been growing since the large explosive (volcanic explosivity index 4) eruption in 2010[30]. This new summit dome has since been partially destroyed by six intermittent explosions between 2012 and 2014[31]. One of these

explosions left a ~200 m-long and up to 40 m-wide open fissure within the dome and an unstable sector within the southern flank of the dome[32], underscoring the link between explosive activity and dome instability at Merapi volcano. A recent explosion on 11 May 2018 was followed by the emergence of a new dome in August 2018.

In total, five large blocks of lava (M-U, M-SA1, M-SA2, M-HA1, and M-HA2; photographs of the blocks are provided in Supplementary Fig. 1) were collected in September 2017 from the summit area of Merapi volcano, ~100 m to the northeast of the active dome in an area where materials were safely accessible. These blocks, extruded in 1902, were selected as representative of the various degrees of visually discernible alteration present. We supplemented these blocks with an additional block collected from deposits of the 2006 eruption (M-2006; Supplementary Fig. 1). The mineral content of the blocks was quantified using X-ray powder diffraction (XRPD) and their microstructure was analysed using a scanning electron microscope (SEM) (see Methods). We measured the connected porosity and permeability of between ten and eleven cylindrical core samples extracted from each of the five main blocks (cores from the same block were all cored in the same orientation), as well as five core samples prepared from the 2006 block (57 core samples in total) (see Methods).

The blocks are characterised by a porphyritic texture comprising phenocrysts of dominantly plagioclase and pyroxene (and high-density oxides) within a crystallised groundmass of plagioclase, K-feldspar, and pyroxene microlites. Backscattered SEM images of each of the blocks are provided as Supplementary Fig. 2. Alteration phases, where present, include alunite, natroalunite, quartz, hematite, cristobalite, gypsum, and unidentifiable amorphous phases (Table 1). The most abundant alteration phases—alunite and natroalunite (Table 1)—are stable over a wide range of temperatures (from room temperature to more than 380 °C) and require acidic, oxidising conditions and a fluid with a high sulphate content[33,34]. We therefore consider that the alteration experienced by these materials was primarily the result of the circulation and cooling of medium- to high-temperature (>200 °C), acidic (pH < 3) fluids.

Block M-U is the least altered and contains no gypsum or alunite-group (aluminium potassium sulphate) minerals (Table 1), but is highly microfractured and the inside of some pores (between 100 and 500 μm in diameter) are coated with cristobalite microcrystals. Block M-SA1 contains small quantities of gypsum and alunite-group minerals (0.5 and 1 wt.%, respectively; Table 1) and is also highly microfractured (especially

---

**Table 1 X-ray powder diffraction (XRPD) analysis showing quantitative bulk mineralogical composition for the five main blocks collected for this study (in wt.%)**

| Mineral | M-U | M-SA1 | M-SA2 | M-HA1 | M-HA2 |
|---|---|---|---|---|---|
| Plagioclase | 54 ± 3 | 47 ± 3 | 38 ± 3 | 38 ± 3 | 19 ± 3 |
| K-Feldspar | 19 ± 3 | 9 ± 3 | 13 ± 3 | 6 ± 3 | 10 ± 3 |
| Clinopyroxene ± orthopyroxene | 16 ± 2 | 13 ± 2 | 14 ± 2 | 11 ± 2 | 8 ± 2 |
| Magnetite | 3 ± 0.5 | 2 ± 0.5 | 2.5 ± 0.5 | <1 ± 0.5 | <1 ± 0.5 |
| Gypsum* | – | 0.5 ± 0.5 | 4 ± 0.5 | 5 ± 0.5 | 6 ± 0.5 |
| K-Na-Alunite* | – | 1 ± 0.5 | 8.5 ± 2 | 11 ± 2 | 24 ± 2 |
| Quartz* | 1 ± 0.5 | 1.5 ± 0.5 | 0.5 ± 0.5 | 1 ± 0.5 | 0.5 ± 0.5 |
| Hematite* | 0.5 ± 0.5 | 2 ± 0.5 | 0.5 ± 0.5 | 3 ± 0.5 | 1 ± 0.5 |
| Cristobalite* | 6 ± 0.5 | – | – | – | 2.5 ± 0.5 |
| Amorphous phases* | – | 24 ± 4 | 19 ± 4 | 25 ± 4 | 28 ± 4 |

An asterisk denotes an alteration phase

in the phenocrysts). The pores are between 100 and 1000 μm in diameter. Block M-SA2 contains more gypsum and alunite-group minerals than block M-SA1 (4 and 8.5 wt.%, respectively; Table 1) and, although microfractures are present, there are qualitatively fewer in M-SA2 than in blocks M-U and M-SA1. The pores within block M-SA2 are between 50 and 300 μm in diameter. Block M-HA1 contains a high quantity of alunite-group minerals (11 wt.%) and gypsum (5 wt.%) (Table 1). The pores within block M-HA1 are large (up to 1000 μm in diameter) and form contorted shapes. Microcracks are present, but are largely confined to large, altered phenocrysts. Block M-HA2 is the most altered and contains high contents of alunite-group minerals and gypsum (24 and 6 wt.%, respectively), as well as other alteration minerals such as hematite and cristobalite (Table 1). The microstructure of block M-HA2 is heterogeneous and pores range from a few tens of microns up to almost 1000 μm, although there are few microfractures. The plagioclase phenocrysts in blocks M-SA1, M-SA2, M-HA1, and M-HA2 are highly altered and often contain fractures and pores that are sealed with alteration minerals, often alunite or natroalunite. Based on the results of our mineralogical and microstructural analyses, we categorise the blocks as: least altered (M-U), slightly altered (M-SA1 and M-SA2), and highly altered (M-HA1 and M-HA2).

The alteration of the blocks to form a sulphur-bearing mineral assemblage comprising natroalunite, alunite, and gypsum (Table 1) is considered here to be the result of fluid-rock interactions following exposure to acid-sulphate fluids[33,34]. This type of alteration is common to the domes and craters of many active volcanoes worldwide, e.g. Merapi volcano[21], Mount Adams, Mount Hood, Mount Rainer, and Mount Shasta (USA)[35,36], Usu volcano (Japan)[37], Soufrière Hills volcano (Montserrat, West Indies)[38], La Soufrière de Guadeloupe (Lesser Antilles)[39], Citlaltépetl volcano (Mexico)[35], Vulcano (Italy)[40], Whakaari volcano (New Zealand)[41,42], and Poás volcano[26,43]. The altered dome materials studied herein are therefore representative for basaltic-andesite and andesite volcanoes worldwide. Importantly, recent geophysical imaging at active volcanoes has shown that the vertical and lateral extent of these hydrothermally altered zones can be on the order of a few hundred metres[20,21].

**Porosity and permeability data**. Permeability as a function of connected porosity is shown in Fig. 1, alongside representative photographs of the 20-mm-diameter samples prepared for the laboratory analyses (data from this study and from Kushnir et al.[44]; see Table 2 for the tabulated dataset). These data show that the porosity and permeability of unaltered dome rock from Merapi can vary from ~0.08 to ~0.28 and from ~$2 \times 10^{-17}$ to ~$1 \times 10^{-11}$ m², respectively (Fig. 1b). We also note that the permeability of the unaltered dome rock increases as connected porosity is increased (indicated by the grey zone in Fig. 1b), in agreement with many published studies for unaltered andesites and basaltic-andesites worldwide[6,45–49].

The porosities and permeabilities of the slightly altered samples (M-SA1, M-SA2), M-2006 (cristobalite alteration), and the cristobalite-bearing samples of Kushnir et al.[44] follow the trend of the unaltered rocks (indicated by the grey zone in Fig. 1b). However, not only are the core samples from the highly altered blocks (M-HA1 and M-HA2) less permeable than their porosity would suggest, but their permeability also varies by up to four orders of magnitude, despite their narrow porosity range. For example, although the difference in porosity between samples M-HA1-10 and M-HA1-2 (samples cored from the same block) is only 0.03, their permeabilities are $2.1 \times 10^{-13}$ and $4.9 \times 10^{-17}$ m², respectively (Fig. 1b; Table 2).

## Discussion

Our data show that the slightly altered samples (M-SA1 and M-SA2) follow the porosity-permeability trend delineated by the unaltered samples (Fig. 1b). However, data from the two highly altered samples (M-HA1 and M-HA2) are characterised by very different porosity-permeability trends (see alteration trajectories indicated by the arrows in Fig. 1b). We interpret this variation to be the result of differences in pore-coating, pore-filling, and microfracture-filling precipitation in the highly altered samples (Fig. 2), which greatly decreases permeability, but does not significantly decrease porosity. This is because, although microfractures provide important flow paths in volcanic rocks[48], they represent only a small volume of the porosity within the rock. Therefore, when these microfractures are sealed or partially sealed (Fig. 2b, e) as minerals precipitate from the circulating hydrothermal fluids, a small decrease in sample porosity can result in considerable permeability reduction. Indeed, volcanic rock samples with similar porosities can be characterised by very different permeabilities, a function of the connectivity of their void space[50]. The difference in porosity-filling alteration is also observable on the sample scale. For example, photographs of the samples of M-HA1 with permeabilities of $2.1 \times 10^{-13}$ and $4.9 \times 10^{-17}$ m² show visible differences in their degrees of alteration (Fig. 1d). Thus, we document that acid-sulphate alteration can reduce the permeability of dome rock by at least four orders of magnitude on the sample lengthscale.

Laboratory measurements of permeability (typically performed on core samples between 10 and 40 mm in diameter) are inherently scale-dependent. For example, they do not account for macroscopic fractures, while we know from field observations that lava domes are highly fractured[51]. Using an effective medium approach, we modelled the equivalent permeability, $k_e$, of a rock mass populated by flow-parallel fractures using the method outlined in Heap and Kennedy[49]:

$$k_e = \frac{(w_{intact} \cdot k_0) + (w_{fracture} \cdot k_f)}{W}, \tag{1}$$

where $k_0$ and $k_f$ are the permeability of the host rock and the fracture permeability, respectively, $w_{intact}$ and $w_{fracture}$ are the width of the host rock and the total fracture width, respectively, and $W$ is the total width (i.e. $w_{intact} + w_{fracture}$). To provide a value for $k_f$ we prepared two additional samples from unaltered block M-U (25 mm in diameter and 25 mm in length). The permeability of these samples was measured using the procedure outlined in the Methods section, after which they were wrapped in electrical tape and loaded diametrically in compression in a servo-controlled uniaxial loadframe until the formation of a through-going tensile fracture (orientated parallel to the direction of fluid flow in the permeability setup). The permeability of the now-fractured samples (i.e. the permeability of samples containing two intact portions separated by a fracture) was then remeasured using the same laboratory procedure (see Methods section). The permeability of the fracture, $k_f$, can be calculated using:

$$k_f = \frac{(A \cdot k_e) - (A_{intact} \cdot k_0)}{A_f}, \tag{2}$$

where $A$ is the cross-sectional area of the sample, $A_{intact}$ is the area of intact material, and $A_f$ is the area of the fracture. If we consider that the fractures are 0.25 mm wide (a reasonable estimate based on measurements made on the fractured samples), then the average fracture permeability for the samples of M-U, calculated using Eq. 2, is $1.5 \times 10^{-10}$ m².

To upscale our laboratory measurements, we considered a lava dome with a length of 100 m that hosts 400 fractures (a fracture density of 4 m⁻¹ is a reasonable estimate for the dome at Merapi

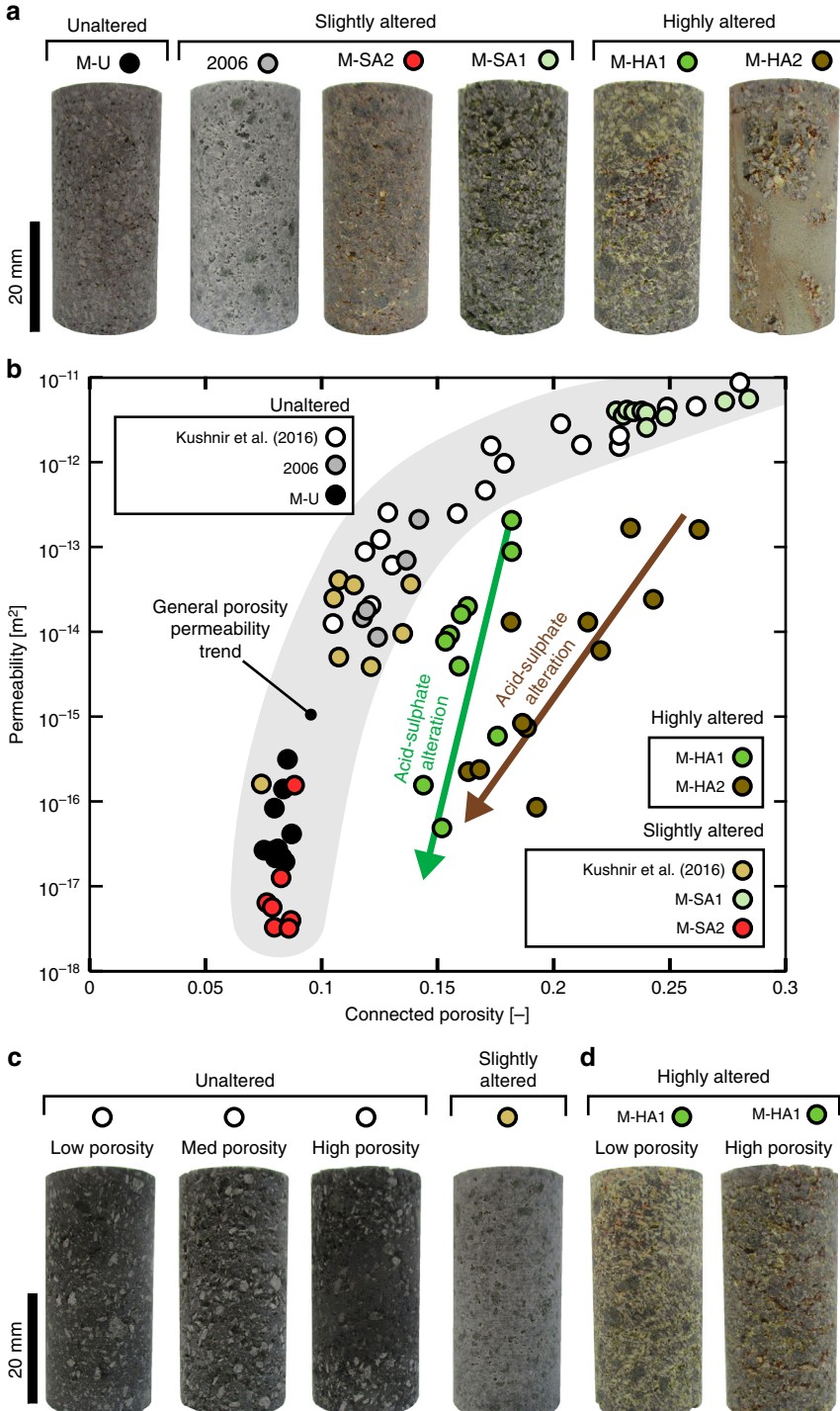

**Fig. 1** Porosity-permeability trends for unaltered and altered dome rocks. **a** Photographs of representative 20-mm diameter core samples prepared from each of the blocks collected for this study. **b** Permeability as a function of connected porosity for dome rocks from Merapi volcano (data from this study and Kushnir et al.[44]). Grey zone shows the general porosity-permeability trend for lavas from Merapi volcano and the arrows show porosity-permeability trajectories for acid-sulphate altered lava dome samples. The experimental error on these measurements is <1% and is therefore within the symbol size. **c** Photographs of representative 20-mm diameter cores from the unaltered and slightly altered (containing cristobalite) samples from Kushnir et al.[44]. **d** Photographs of two 20-mm-diameter core samples prepared from block M-HA1 that preserve different degrees of alteration

volcano[32]). We assumed the permeability of these fractures to be the same as determined in our above-described laboratory experiments (i.e. $1.5 \times 10^{-10}$ m$^2$) and a fracture width of 2 cm (a reasonable estimate for the fractures within the dome at Merapi volcano). We considered three scenarios: an unaltered dome with a host rock permeability of $1.0 \times 10^{-13}$ m$^2$ in which all fractures

are open, a moderately altered dome with a host rock permeability of $1.0 \times 10^{-15}$ m$^2$ in which 50% of the fractures are sealed, and a highly altered dome with a host rock permeability of $1.0 \times 10^{-17}$ m$^2$ in which 99% of the fractures are sealed. We assumed that a sealed fracture has a permeability of zero. The equivalent permeability of the fractured lava dome for these three scenarios,

**Table 2 The connected porosity and permeability for the samples prepared from the blocks collected for this study. Permeability was measured under a confining pressure of 1 MPa (see Methods section for details). The experimental error on these measurements is <1%**

| Sample | Connected porosity | Permeability (m²) |
|---|---|---|
| M-SA1-1 | 0.283 | $4.13 \times 10^{-12}$ |
| M-SA1-2 | 0.230 | $3.61 \times 10^{-12}$ |
| M-SA1-3 | 0.248 | $3.50 \times 10^{-12}$ |
| M-SA1-4 | 0.227 | $4.02 \times 10^{-12}$ |
| M-SA1-5 | 0.235 | $3.92 \times 10^{-12}$ |
| M-SA1-6 | 0.240 | $2.59 \times 10^{-12}$ |
| M-SA1-7 | 0.232 | $4.12 \times 10^{-12}$ |
| M-SA1-8 | 0.240 | $3.88 \times 10^{-12}$ |
| M-SA1-9 | 0.284 | $5.66 \times 10^{-12}$ |
| M-SA1-10 | 0.238 | $4.00 \times 10^{-12}$ |
| M-SA1-11 | 0.271 | $5.27 \times 10^{-12}$ |
| M-SA2-1 | 0.086 | $3.18 \times 10^{-18}$ |
| M-SA2-2 | 0.082 | $1.26 \times 10^{-17}$ |
| M-SA2-3 | 0.087 | $4.00 \times 10^{-18}$ |
| M-SA2-4 | 0.080 | $3.23 \times 10^{-18}$ |
| M-SA2-5 | 0.079 | $5.84 \times 10^{-18}$ |
| M-SA2-6 | 0.084 | $2.01 \times 10^{-17}$ |
| M-SA2-7 | 0.077 | $6.15 \times 10^{-18}$ |
| M-SA2-8 | 0.088 | $1.58 \times 10^{-16}$ |
| M-SA2-9 | 0.078 | $5.70 \times 10^{-18}$ |
| M-SA2-10 | 0.083 | $2.08 \times 10^{-17}$ |
| M-HA2-1 | 0.185 | $8.94 \times 10^{-16}$ |
| M-HA2-2 | 0.182 | $1.31 \times 10^{-14}$ |
| M-HA2-3 | 0.192 | $8.71 \times 10^{-17}$ |
| M-HA2-4 | 0.215 | $1.34 \times 10^{-14}$ |
| M-HA2-5 | 0.233 | $1.75 \times 10^{-13}$ |
| M-HA2-6 | 0.220 | $6.62 \times 10^{-15}$ |
| M-HA2-7 | 0.188 | $7.16 \times 10^{-16}$ |
| M-HA2-8 | 0.163 | $2.31 \times 10^{-16}$ |
| M-HA2-9 | 0.242 | $2.41 \times 10^{-14}$ |
| M-HA2-10 | 0.263 | $1.64 \times 10^{-13}$ |
| M-HA2-11 | 0.168 | $2.37 \times 10^{-16}$ |
| M-HA1-1 | 0.159 | $4.07 \times 10^{-15}$ |
| M-HA1-2 | 0.152 | $4.86 \times 10^{-17}$ |
| M-HA1-3 | 0.176 | $5.99 \times 10^{-16}$ |
| M-HA1-4 | 0.154 | $7.81 \times 10^{-15}$ |
| M-HA1-5 | 0.182 | $8.93 \times 10^{-14}$ |
| M-HA1-6 | 0.144 | $1.54 \times 10^{-16}$ |
| M-HA1-7 | 0.155 | $9.35 \times 10^{-15}$ |
| M-HA1-8 | 0.160 | $1.56 \times 10^{-14}$ |
| M-HA1-9 | 0.162 | $2.05 \times 10^{-14}$ |
| M-HA1-10 | 0.182 | $2.11 \times 10^{-13}$ |
| M-U-1 | 0.081 | $2.73 \times 10^{-17}$ |
| M-U -2 | 0.087 | $4.16 \times 10^{-17}$ |
| M-U -3 | 0.083 | $1.41 \times 10^{-16}$ |
| M-U -4 | 0.080 | $8.50 \times 10^{-17}$ |
| M-U -5 | 0.080 | $2.47 \times 10^{-17}$ |
| M-U -6 | 0.085 | $3.18 \times 10^{-16}$ |
| M-U -7 | 0.083 | $2.17 \times 10^{-17}$ |
| M-U -8 | 0.079 | $2.54 \times 10^{-17}$ |
| M-U -9 | 0.075 | $2.70 \times 10^{-17}$ |
| M-U -10 | 0.080 | $2.23 \times 10^{-17}$ |
| M-2006-1 | 0.119 | $1.84 \times 10^{-14}$ |
| M-2006-2 | 0.136 | $6.99 \times 10^{-14}$ |
| M-2006-3 | 0.124 | $8.77 \times 10^{-15}$ |
| M-2006-4 | 0.142 | $2.14 \times 10^{-13}$ |
| M-2006-5 | 0.118 | $1.49 \times 10^{-14}$ |

however, the permeability of the dome is reduced to $9.2 \times 10^{-18}$ m², highlighting the importance of few, or even isolated, fractures in maintaining the high dome permeability required for efficient outgassing of the underlying magma-filled conduit.

It is important to assess how a reduction in the equivalent permeability of a dome from $10^{-11}$ to $10^{-13}$ m² (i.e. unaltered to highly altered) will influence pore pressure. To do so, we numerically modelled gas loss using a 2D finite element approach in COMSOL Multiphysics V4.3 in which we combined the continuity equation and Darcy's law, deriving a partial differential equation that was solved for pressure[1]. The model was split into three domains: the magma-filled conduit, the edifice, and the lava dome (see Fig. 3a). To assess the role of dome permeability, we fixed the equivalent permeability of the magma-filled conduit and edifice at, respectively, $10^{-10}$ and $10^{-13}$ m², and varied the equivalent permeability of the lava dome from $10^{-11}$ to $10^{-13}$ m² (the results of additional simulations are provided in Supplementary Fig. 3). For these three scenarios, corresponding to the unaltered, slightly altered, and highly altered dome scenarios described above, the maximum overpressure beneath the dome increased from 11.96, to 21.83, and, finally, to 27.14 MPa as dome permeability decreased from $10^{-11}$ to $10^{-13}$ m² (Fig. 3; tabulated results can be found in Supplementary Table 1). Additional simulations show that a similar pattern of pressure augmentation is seen for domes with different heights (50, 100, and 200 m) and that the magnitude of the overpressure within and beneath the dome depends on the edifice permeability (higher overpressures are possible for lower edifice permeabilities; see Supplementary Fig. 3). We therefore conclude that progressive permeability reduction due to the hydrothermal alteration of a lava dome can significantly increase the pore overpressure within and beneath the dome, leaving the system prone to explosive behaviour.

In a next step we assess whether the overpressures predicted by our modelling (Fig. 3) are capable of fragmenting rock and magma. The fragmentation criterion, derived from the stress distribution surrounding isolated spherical pores[52], has been shown to well describe the available experimental data for the fragmentation threshold, $P_{th}$, for volcanic rocks and magmas:

$$P_{th} = \frac{2S(1-\phi)}{3\phi\sqrt{\phi^{-1/3}-1}},\qquad(3)$$

where $S$ and $\phi$ are the effective tensile strength and porosity of the material, respectively. Using a value of $S$ that well describes experimental data for andesites from Volcán de Colima (Mexico)[53], Eq. (3) suggests that the maximum overpressure modelled beneath an unaltered dome characterised by a permeability of $10^{-11}$ m² (11.96 MPa; Fig. 3b) is capable of fragmenting material with a porosity of ~0.16. An increase in overpressure to 27.14 MPa (i.e. the highly altered dome scenario, Fig. 3d) allows for the fragmentation of material with a porosity as low as ~0.05. Porosity values for the rock samples measured herein vary from 0.08 to 0.28 (Table 2), and laboratory porosity values for historical dome samples vary significantly, from ~0.01 up to ~0.5[44,54]. Electromagnetic tomography at Merapi volcano has yielded porosity estimates of 0.05–0.1[55]. An increase in overpressure from 11.96 to 27.14 MPa is therefore sufficient to fragment the vast majority of the rocks and magma within and beneath the dome at Merapi volcano. Further, if hydrothermal alteration also reduces the effective tensile strength of the dome materials[56–59], the fragmentation threshold of a rock with a given porosity will be lowered. We note that, even if the permeability of the edifice is lowered to $10^{-12}$ m², the overpressures generated in our highly altered dome scenario are still capable of fragmenting the majority of the rocks and magma within and beneath the dome (see Supplementary Fig. 3).

using Eq. 1, is $1.2 \times 10^{-11}$, $6.0 \times 10^{-12}$, and $1.2 \times 10^{-13}$ m², respectively. Interestingly, reducing the host rock permeability by four orders of magnitude and sealing 99% of the fractures only reduces the equivalent permeability of the dome by about two orders of magnitude. When 100% of the fractures are sealed,

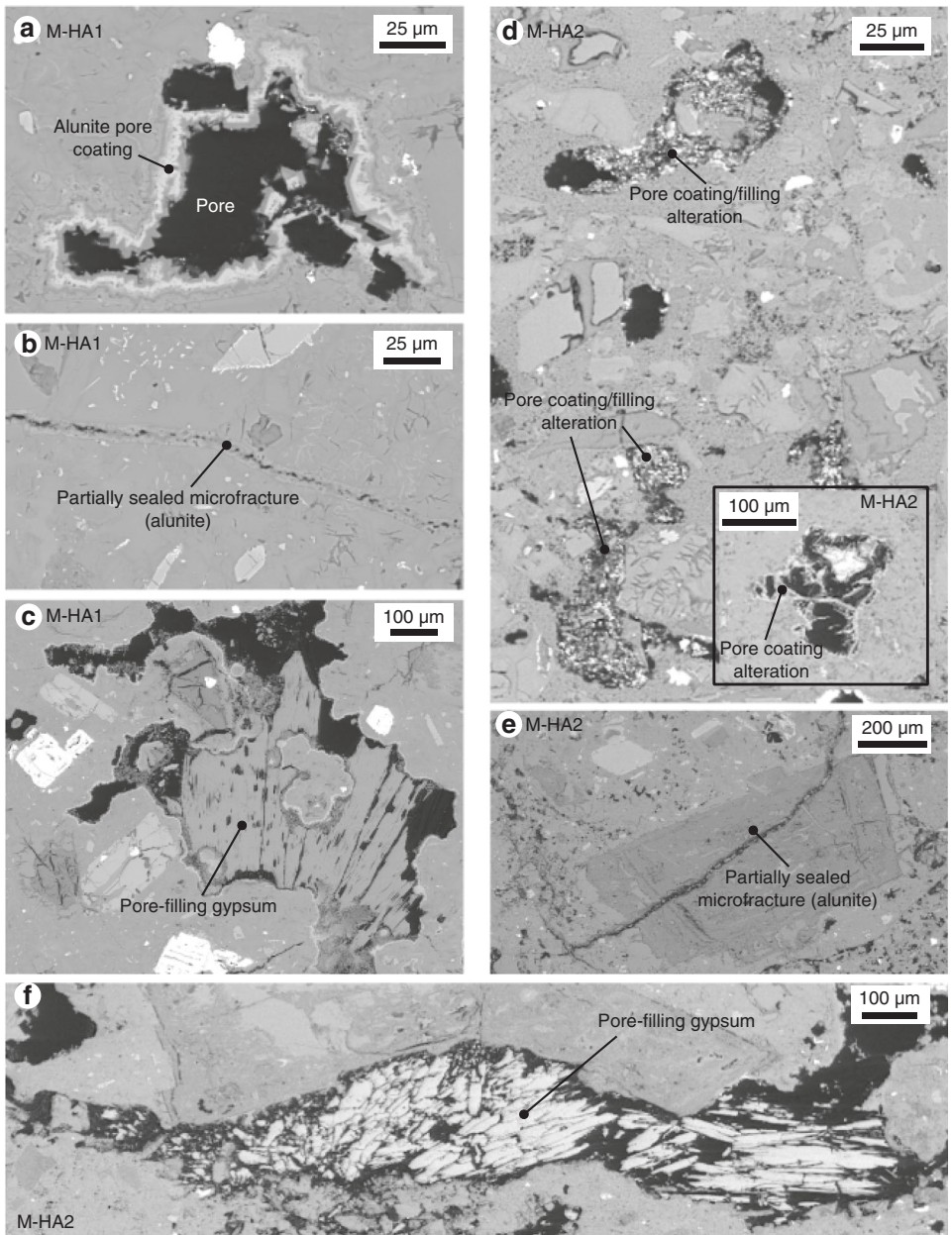

**Fig. 2** Porosity-filling alteration. Backscattered scanning electron microscope images showing **a** a pore that is partly filled with alunite in block M-HA1, **b** a fracture partially sealed by alunite precipitation in block M-HA1, **c** a pore filled with gypsum and coated with alunite in block M-HA1, **d** pore-coating and pore-filling alteration in block M-HA2, **e** a fracture partially sealed by alunite precipitation in block M-HA2, **f** a pore filled with gypsum in block M-HA2

A final consideration is the time required to produce wide-spread alteration of a lava dome. At Merapi volcano, for example, sequential images of the lava dome (taken using a drone) show that secondary mineral deposition at the surface, which we consider here to be also associated with significant alteration at depth, can develop in less than three years[60]. Rapid reduction in dome permeability through acid-sulphate alteration of the 2010 lava dome could therefore explain the six volatile-driven dome explosions between 2012 and 2014 and the recent explosion in May 2018. If true, the six explosions within two years suggest that acid-sulphate alteration sufficiently reduced permeability within a timescale of just months to years, and that the process occurred repeatedly. Using time-lapse photography, we can test the hypothesis that a short term sealing process led to a decrease in permeability and an increase in pore pressure prior to the recent May 2018 explosion. Time-lapse photography of the May 11 2018

explosion (Fig. 4) highlights that the focussed outgassing at the dome rim (Fig. 4a) stopped on May 5 (Fig. 4b) and that there was no visible outgassing until the large explosion on May 11 (Fig. 4c) (more images are available in Supplementary Fig. 4). Following the explosion, diffuse outgassing was observed from the dome summit. We interpret the 2018 explosion as a result of the cessation of outgassing caused by hydrothermal sealing, as shown in the accompanying schematic diagrams in Fig. 4. Although the appearance of outgassing can depend on environmental factors, such as air temperature and pressure, we note that the presence and absence of outgassing in the run-up to the May 11 explosion did not depend on time-of-day or changing weather conditions. Although high temporal resolution $SO_2$ flux data are currently unpublished for Merapi volcano, a reduction in pre-eruptive $SO_2$ flux has been observed at, for example, Galeras volcano[18], Sou-frière Hills volcano (Montserrat)[23], Popocatépetl volcano

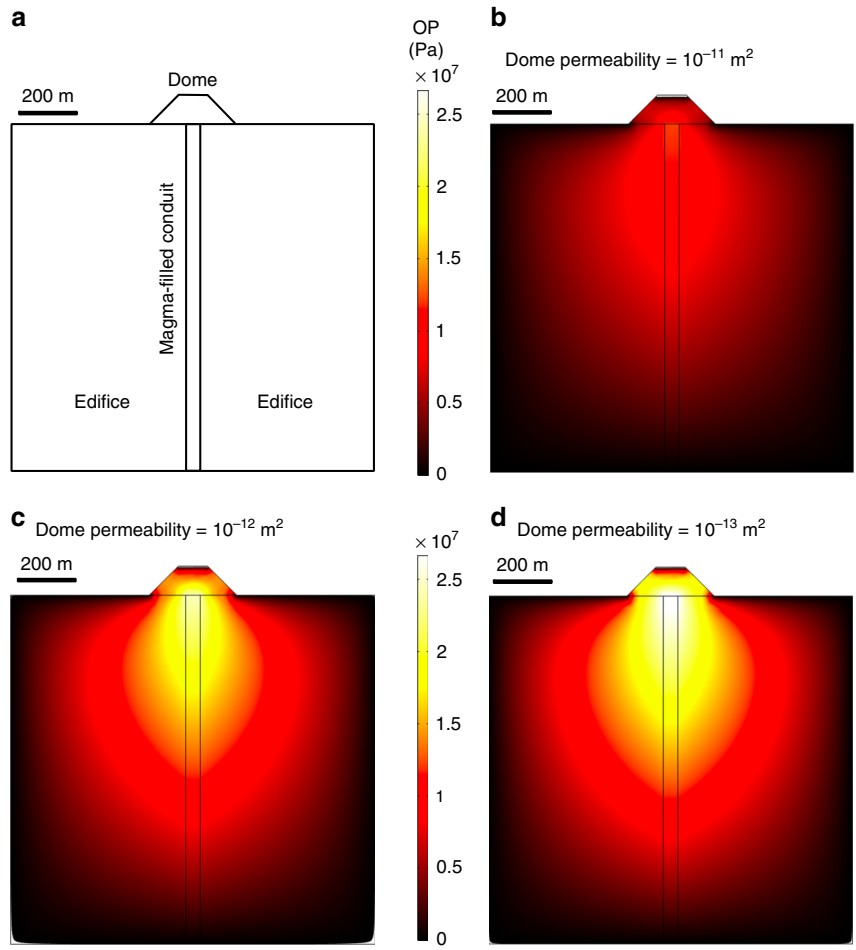

**Fig. 3** Pore pressure augmentation within and beneath a lava dome. **a** Model setup. **b–d** Numerical models showing the magnitude and distribution of pore overpressure (white and yellow represent high overpressure and red and dark red represent low overpressure) for domes with different equivalent permeabilities (ranging from $10^{-11}$ to $10^{-13}$ m²). The models in panels **b**, **c**, and **d** are designed to represent an unaltered, moderately altered, and highly altered dome, respectively. OP overpressure

(Mexico)[61], and Poás volcano[26], lending support to the mechanism outlined in Fig. 4. Ongoing alteration and permeability reduction at Merapi volcano may therefore offer an explanation for the frequent and erratic explosive dome outbursts that are not associated with magma recharge events from depth[62,63], as was the case for the 2010 event. This type of intermittent explosive activity can destabilise an already-unstable lava dome, like the dome at Merapi volcano[60], which could in turn trigger a large flank failure and a consequent larger eruption involving the formation of potentially devastating pyroclastic density currents.

We conclude that acid-sulphate alteration can rapidly, over months to years, reduce the permeability of lava domes worldwide, promoting pore pressure increases and irregular explosive volcanic behaviour. We further note that hydrothermal alteration typically weakens volcanic rock[19,56–58,64] and that such weakening could reduce the stability of the dome and further increase the likelihood of unexpected dome explosions and associated hazardous pyroclastic density currents[4,19,65–67]. On the basis of our findings, mapping the extent and evolution of hydrothermal alteration at active lava domes using geophysical methods such as electrical[20,21,68,69] and muon tomography[70–72], spectroscopic methods such as visible and infrared spectroscopy[36,73], and gas monitoring[26] emerge as an important tools to help anticipate dome explosions at otherwise unpredictable dome-forming volcanoes.

## Methods

**X-ray powder diffraction**. The mineral content of the five blocks was quantified using X-ray powder diffraction (XRPD) on powdered offcuts of the experimental samples. Powdered samples were ground for 8 min with 10 ml of isopropyl alcohol in a McCrone Micronising Mill using agate cylinder elements. The XRPD analyses were performed on powder mounts using a PW 1800 X-ray diffractometer (CuKα, graphite monochromator, 10 mm automatic divergence slit, step-scan 0.02° 2θ increments per second, counting time one second per increment, 40 mA, 40 kV). The mineral phases in the whole rock powders were quantified using the Rietveld refinement program BGMN[74]. We also separated <2 μm fractions by gravitational settling and prepared oriented mounts for X-ray diffraction analysis, but no clay minerals were found.

**Microstructural analysis**. The microstructure and alteration of each of the five blocks was investigated on thin sections prepared from offcuts of the experimental samples using a Tescan Vega 2 XMU scanning electron microscope (SEM).

**Porosity and permeability**. Between ten and eleven cylindrical core samples (20 mm in diameter and nominally 40 mm in length) were prepared from each of the five main blocks, and five core samples from the 2006 block. The cylindrical core samples were then dried in a vacuum oven at 40 °C for at least 48 h. The connected porosities of the samples were then calculated using the bulk sample volume (determined using the sample dimensions) and the connected skeletal volume determined using a helium pycnometer. Permeabilities were measured using a nitrogen gas permeameter[49,75] under ambient laboratory temperature. All measurements were conducted in a pressure vessel under a confining pressure of 1 MPa to ensure that the pore pressure never exceeded the confining pressure and that the pore fluid could not travel between the sample edge and the rubber jacket. Samples were left at the confining pressure for 1 h to ensure microstructural equilibration. Permeability was measured using the steady-state flow method (for high-permeability samples) or the transient pulse-decay

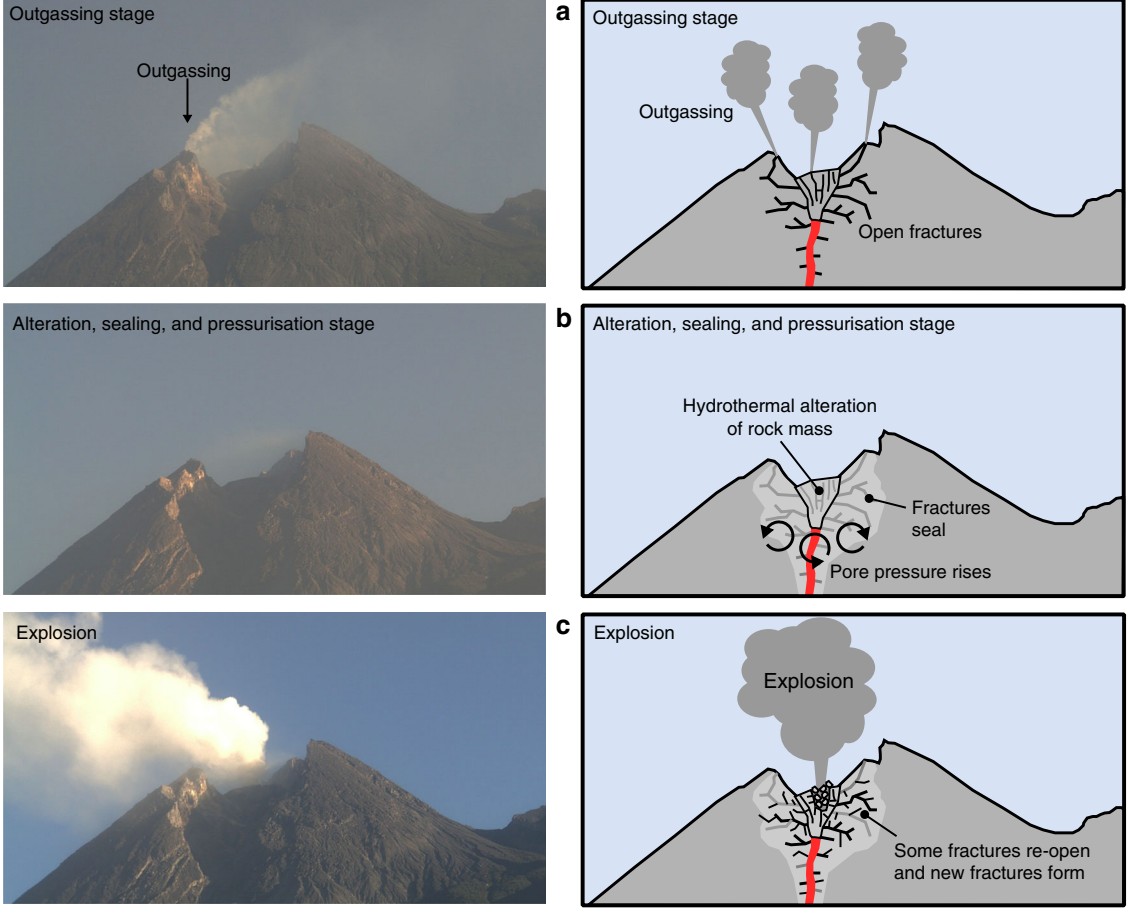

**Fig. 4** Explosive behaviour at Merapi volcano thought to be prompted by hydrothermal alteration. **a** The outgassing stage. Photograph of Merapi on May 3 showing focused outgassing on the dome rim. We interpret this as the result of outgassing through open fractures. **b** Alteration, sealing, and pressurisation stage. Focussed outgassing stopped on May 5 (the photograph shown here is from May 7). We interpret this as the result of the hydrothermal alteration of the dome (alteration of the rock mass and sealing of fractures). This causes pore pressure to rise beneath the dome. **c** Explosion. After 5 days of no to little outgassing, an explosion occurred on May 11. The explosion is interpreted as a consequence of the pore pressure augmentation beneath the dome. Some fractures are re-opened and new fractures form allowing for passive outgassing following the explosion (as shown in Supplementary Fig. 4)

method (for low-permeability samples). For the steady-state flow measurements, volumetric flow rate measurements (using a gas flowmeter) were collected for several pore pressure gradients (monitored using a pressure transducer) to determine permeability using Darcy's law and to check for ancillary corrections such as the Forchheimer and Klinkenberg corrections. For the pulse-decay measurements, we determined permeability, and checked for the above-mentioned corrections, using the decay of a pore pressure gradient (monitored using a pressure transducer). More details on these methods of permeability determination can be found in Heap et al.[76].

### Data availability
The data collected for this study are available in Tables 1, 2.

### Code availability
COMSOL Multiphysics V4.3 is a commercially available physics package (https://www.comsol.com/).

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

## Acknowledgements

This study was partly supported by VOLTAGE (a project funded by the Research Council of Sweden), by the Swedish Center for Hazard and Disaster Sciences (CNDS), by Volcapse (a project funded by the European Research Council under the European Union's H2020 Programme/ERC consolidator), and through a scholarship grant from the Deutscher Akademischer Austauschdienst (DAAD; Germany; reference number 91525854). We also thank Hanik Humaida (BPPTKG, Yogyakarta).

## Author contributions

M.H. led the project and wrote the manuscript. Fieldwork and sample collection were carried out by V.T., N.S., H.D., and F.D. M.H. performed the laboratory measurements of porosity and permeability, with help from A.K. A.K. performed the SEM analyses. A.G. performed the XRPD measurements and analysed the data. A.C. and J.N. performed the numerical modelling. T.W. provided the time-lapse photography. All authors contributed to the writing of the manuscript.

## Competing interests

The authors declare no competing interests.
