## [Peer Review File · Nature Communications]

Reviewers' comments:

Reviewer #1 (Remarks to the Author):

General remarks

In this study the authors suggest that hydrothermal alteration of lava dome rock can lead to explosive activity. I have no problem with this idea, which has been examined by a number of previous studies. However, I believe that the authors have not clearly demonstrated the proper role of hydrothermal alteration when looking at the larger picture. It is likely that a range of processes contribute to explosive activity. These include purely magmatic processes which can contribute to overpressure, and dynamic processes which in large part determine the size and distribution of fracture networks. I also feel that the authors need to do a better job of applying the large range of permeability seen for some individual samples to better comprehend the permeability of the system as a whole. This study has the potential to provide new insight into the lateral and vertical and time-varying permeability distribution of a lava dome system. But to accomplish this, further work and analysis is required.

In the 1990's, Fink and Griffiths showed that for extruding lava domes, both the fracture density and the scale of the fractures are strongly influenced by a balance of the advection time vs. the solidification time. This must be a first-order control upon the distribution of fractures within an actively growing lava dome.

Once a lava dome is emplaced, there must be a balance between magmatic processes such as cooling and crystallization, and hydrothermal processes such as alteration and sealing. Both can lead to pressurization and explosive activity. The trick is understanding this balance in space and time, hence understanding the underlying processes which generate explosive activity. Fractures can be sealed by hydrothermal processes, while new fractures may open from pressure gradients and deeper influences. So there must be a balance here as well. The extent and scale of fracturing are clearly important in this regard.

A key result from Figure 2 is that the two altered samples show a large range of permeability compared to the unaltered and less altered samples. In fact, a single altered sample appears to have close to four orders of magnitude difference in permeability, i.e., there is no characteristic permeability for these samples. What does this mean? What are the implications for individual samples from a lava dome, and what are the implications for a lava dome as a whole?

Figure 2 also shows some previous data from Kushnir et al. 2016. For slightly altered materials, there is one sample (light green) with comparatively high permeability and one sample (brown) with extremely low permeabilities. How can similarly altered samples have such different permeabilities?

Specific comments

Line 42: Please explain what you mean by “steam-driven” explosive activity. What is the source of the steam? Meteoric water? Magmatic water? A combination of the two?

Line 51: Please provide some further explanation regarding the “near-ubiquity” of alteration at lava domes. At what scales does the alteration occur? How extensive is it? How is it distributed across the surface of a lava dome and within its interior?

Lines 78-81: I do not see the utility of this figure. The distinctions among the six photos are not shown, nor are they clear to the reader. In the text (lines 105-106), it is stated that the plagioclase crystals are altered with alteration minerals, but the reader cannot clearly see these features.

Lines 192-193: Please justify this assumption that the width and permeability of the Merapi fractures are the same as in your experiments. Why should this necessarily be so?

Lines 200-216: You state in lines 200-205 that progressive sealing to 99% changes the dome permeability by two orders of magnitude. The last 1% of sealing causes another two-order magnitude decline in permeability. In your modeling, however, the large increases in overpressure happen during the first stages of sealing, i.e., at comparatively high permeability. At low permeabilities, the overpressures are essentially constant and do not change. Please explain this apparent paradox.

Reviewer #2 (Remarks to the Author):

Review of “Hydrothermal Alteration Leads to Explosive Volcanic Behaviour” by M.J. Heap et al.

This study explores hydrothermal sealing of lava domes as a mechanism leading to explosive volcanic eruptions based on the case of Merapi volcano, Indonesia. The topic is an important one given the hazardous nature of this volcano and these types of eruptions. The approach of the study is innovative and combines petrologic observations, quantitative measurements of connected porosity and permeability, and modelling in order to explain observations of volcanic behavior – an elegant and rigorous combination. The manuscript is well written and the data are robust, though based on a relatively small number of samples (which is understandable considering how difficult and dangerous sample collection is at Merapi – the importance of these samples should not be understated). The arguments are generally reasonable and convincing, i.e. that alteration of lava domes leads to decreased porosity and that this could impede gas flow. Hydrothermal sealing is undoubtedly an important process in some eruptions, and this is a point that is worth stressing as it is generally underappreciated in the literature. However, there are number of important issues that need to be addressed. Specifically, more attention needs to be paid to the conditions of the observed alteration and whether this is consistent with sealing of high temperature gas conduits through the dome. Also, the configuration of the model presented in figure 2 does not seem reasonable as the dome is apparently modeled to completely cap the conduit and the surrounding edifice. In reality, lava domes (Merapi is an excellent example) are confined to the conduit area. The authors also argue that the “steam-driven” eruptions at Merapi in 2012, 2014, and 2018 were due to sealing of the dome, however they present no evidence that escape of magmatic gases was inhibited. Therefore, though the manuscript does make a convincing argument that relatively subtle alteration is an important process in decreasing permeability lava domes (an important and provocative finding in itself), it is not fully convincing that this alteration of the dome is in reality responsible for blocking the escape of magmatic gases at Merapi. If these issues can be addressed, I feel that this study would make a good contribution to Nature Communications.

Major issues are detailed below, and specific comments are provided in the annotated pdf.

Major issues:

1. More details are needed on the conditions of hydrothermal alteration observed in the rock samples. Is this alteration consistent with high temperature alteration by gases in the pathways along which they are travelling, or more consistent with low T alteration/weathering by meteoric fluids with a component of dissolved gas? The former case would be consistent with the proposed

model and implications of the manuscript, implying sealing of the principal gas pathways. The latter would imply that the sealing is only affecting the peripheral dome away from the high T gas pathways, which may not be that important for escape of gases (as implied by the results stated on l. 200-202).

2. Modelling of pore pressures and gas velocity – sensitivity analysis. It is important to conduct sensitivity analysis of the model by varying some of the key variables such as conduit and edifice permeabilities, and provide more information on why the values used were chosen.

3. Configuration of pore pressure model. The geometry of the model is not realistic. As shown in figure 4, the conduit and edifice are completely capped by a dome, with lateral dimensions of 1300m. In nature, lava domes (at least the ones pertinent to this study) have lateral dimensions of the same order of the dimensions of the conduit, which in this case is narrow (50m?). Details of the model geometry are not given in the text or Supplemental Information. Probably, if the dome has more reasonable dimensions the gas would simply flow around the dome through the more permeable edifice.

4. A key argument of the paper is that the “steam-driven” eruptions at Merapi in 2012, 2014, and 2018 are evidence that the hydrothermal sealing process is significant in nature and leads to eruption. However, no evidence is given that magmatic gas flow through the dome was impeded prior to these eruptions (this is a relatively well-monitored volcano – there should be gas data to support the idea). Furthermore, if accumulation of magmatic gases resulted in pore pressure increase and dome failure then these are not steam-driven eruptions but are in fact driven by magmatic gas.

5. More attention is needed on error assessment. Please report errors in Table 2 data and show error bars in Figure 2.

Maarten de Moor

19 January 2019

Reviewer #3 (Remarks to the Author):

General comments

The submitted manuscript presents an experimental study, integrated by numerical modelling, on the effect of hydrothermal alteration on lava domes as one process controlling their explosive behavior.

The authors performed mineralogical and petrophysical characterization of series of rock samples from Merapi volcano, representative of basaltic-andesitic/andesitic volcano-type of rocks, and differently altered by acid-sulfate alteration. Laboratory measurements indicate that (moderate to highly) altered dome rocks from Merapi, not only are less permeable than their porosity would suggest, but their permeability also varies by up to four orders of magnitude ($\sim 10^{-13}$ to $\sim 10^{-17}$ m²) within a narrow porosity range (~ 0.15 to ~ 0.25). The permeability of artificially fractured specimens was also estimated to be in the order of $\sim 10^{-10}$ m². To upscale the laboratory measurements authors simulated, by means of a numerical modeling, a ten-meter-long dome with unaltered to moderately and altered rocks, having fractures differently sealed (from complete open system to 99% sealed fractures). Numerical results show that intense alteration can reduce the rocks/fractures permeability of a dome by two orders of magnitude switching the outgassing regime from “open” to “partially closed”, thus favoring pore pressurization within and beneath the dome, and priming the system for explosive behaviour.

Results are significant, and shed a quantitative light on the very important effect of the hydrothermal alteration as a priming factor for volcanic explosions at volcanic domes, therefore the study is definitely worthy of publication. I recommend publication after that the following comments (minor) have been suitably addressed.

Cristian Montanaro

Main comments

Starting from the manuscript Title... The authors do only explore how acid-sulfate alteration affects dome rocks from the basaltic-andesitic and andesitic types. It is true that those are the most widely distributed (and erupting) once, though it would be worthy to discuss how similar alteration processes may affect different dome rock compositions (e.g. dacite, rhyodacite or rhyolite?) to put

the study's results in a broader context. If not I think a title like "Hydrothermal alteration of a andesitic dome leads to explosive volcanic behavior" would be more appropriate.

In the Results, line 123-126, the sentence is very long and not clear in some points. I suggest breaking it in two, and specify in line 124 that you talk about the trend observed for unaltered rocks.

"respectively. In particular it can be observed how the permeability of unaltered dome samples"

Additionally, the deviation in permeability/porosity of altered rocks, in respect to the general trend, should be discussed briefly to highlight the decrease of permeability due to pore-filling minerals, and likely to the formation of complex connected pore network.

In Line 143, it is stated that the slightly altered material has permeability/porosity following the main trend of unaltered rocks, however, they seems already to show a slight shift down in permeability (likely due to the pore-filling effect), for samples having similar porosity range. In case, I think this should be briefly discussed.

In Line 144, please state that you are talking about "Highly altered material (M-HA1 and M-HA2)"

In Lines 154-156, here should also be discussed how the formation of very complex, and tortuous, pore network may influence/reduce samples permeability having same/similar porosity (e.g. Colombier, M., Wadsworth, F.B., Gurioli, L., Scheu, B., Kueppers, U., Di Muro, A., Dingwell, D.B., 2017. The evolution of pore connectivity in volcanic rocks. *Earth Planet. Sci. Lett.* 462, 99–109; Kushnir, A.R., Martel, C., Bourdier, J.L., Heap, M.J., Reuschlé, T., Erdmann, S. Cholik, N., 2016. Probing permeability and microstructure: unravelling the role of a low permeability dome on the explosivity of Merapi (Indonesia). *J. Volcanol. Geotherm. Res.* 316, 56–71.)

An interesting point is that alteration leads to the formation of micro-pores (tens of micron), which can have quite an important role in case of steam-driven explosive events where liquid water is filling dome rocks (likely to occur, especially in rainy regions). Indeed, water explosivity may be enhanced by the micro pores structure, as theoretically postulated (Thiéry, R., Looock, S., & Mercury, L. (2010). Explosive properties of superheated aqueous solutions in volcanic and hydrothermal systems. In *Metastable Systems under Pressure* (pp. 293–310), and qualitatively observed in experiments (Montanaro, C., Scheu, B., Mayer, K., Orsi, G., Moretti, R., Isaia, R., & Dingwell, D. B. (2016). Experimental investigations on the explosivity of steam-driven eruptions: a case study of

Solfatara volcano (Campi Flegrei). *Journal of Geophysical Research: Solid Earth*, 121(11), 7996–8014. <https://doi.org/10.1002/2016JB013273>)

Maybe this point can be briefly discussed here, and recalled later when discussing potential effect of explosive events in destabilizing domes: for e.g. more energetic explosive event may favor larger removal of dome material.

Lines 213-219: Once a certain overpressure is reached, rock porosity play a further important role in i) storage the gas potential energy, as well as in ii) determining the energy/pressure threshold needed for starting the fragmentation (e.g. Scheu, B., Kueppers, U., Mueller, S., Spieler, O., & Dingwell, D. B. (2008). Experimental volcanology on eruptive products of Unzen volcano. *Journal of Volcanology and Geothermal Research*, 175(1–2), 110–119.

<https://doi.org/10.1016/j.jvolgeores.2008.03.023>; Richard, D., Scheu, B., Mueller, S. P., Spieler,

O., & Dingwell, D. B. (2013). Outgassing: Influence on speed of magma fragmentation. *Journal of Geophysical Research: Solid Earth*, 118(3), 862–877. <https://doi.org/10.1002/jgrb.50080>).

Though the model does not account for this parameter, I think is interesting to discuss (later for e.g. between lines 228-243) the effect of rocks porosity considering those measured from Merapi's samples. For e.g. for dacitic dome rocks having porosities in the range 0.15 to 0.25 an overpressure >10 and <20 MPa is needed for fragmentation, though rock strength weakening due to alteration may reduce the fragmentation threshold (see next comment). Moreover, and considering that permeability determines whether the expanding fluid may either fragment the surrounding rocks (permeability <~10-12m²) or escape from it via effective outgassing (permeability >~10-12 m²; Richard et al., 2013), at Merapi only highly altered rocks (in the 0.15 to 0.25 porosity range) have permeability lower enough (<10-13m²) to favor fragmentation. I think these arguments may further strengthen the results found in this study.

Lines 246-249: I think is worthy to add references about the effect of acid-sulfate alteration in decreasing rocks strength as already observed in other volcanic terrains (e.g. Solfatara in Campi Flegrei, Italy – e.g. Mayer, K., Scheu, B., Montanaro, C., Yilmaz, T. I., Isaia, R., Aßbichler, D., & Dingwell, D. B. (2016). Hydrothermal alteration of surficial rocks at Solfatara (Campi Flegrei): Petrophysical properties and implications for phreatic eruption processes. *Journal of Volcanology and Geothermal Research*, 320, 128–143. <https://doi.org/10.1016/j.jvolgeores.2016.04.020>) or other andesitic domes (e.g. Santiaguito dome complex in Guatemala, e.g. Ball, J. L., Calder, E. S., Hubbard, B. E., & Bernstein, M. L. (2013). An assessment of hydrothermal alteration in the Santiaguito lava dome complex, Guatemala: Implications for dome collapse hazards. *Bulletin of Volcanology*, 75(1), 1–18. <https://doi.org/10.1007/s00445-012-0676-z>), since the jointed effect of permeability and strength reduction may i) lower the rock fragmentation threshold (thus reducing overpressures needed for explosive failures) and favor (minor to large) flank insatiability.

Minor corrections

Line 26: "rock samples"

Line 71: "supplemented"

Line 85-86: "Alteration phases present in the samples included"

Lines 112-117: "domes, and craters of many..." (otherwise Vulcano and Whakaari volcano should be cut from the list)

Lines 120: "alongside with"

Line 150: "manifested as"

Line 177: "from the unaltered block"

Line 182: "remeasured"... by using a permeameter? In case just specify and add that you are measuring permeability of fractures and connected pore, and after the permeability due to fractures can be estimated using the formula 2

Line 188: "equivalent" (?)

Figure 2: Black circles are not in the plot. Are they missing? Covered by other data symbol? Or the color of some of the data symbols is wrong? Please check.

Figure 4: can the $\times 10^7$ in the OP scale bar be removed?

Reviewer #1

General comments

In this study the authors suggest that hydrothermal alteration of lava dome rock can lead to explosive activity. I have no problem with this idea, which has been examined by a number of previous studies. However, I believe that the authors have not clearly demonstrated the proper role of hydrothermal alteration when looking at the larger picture. It is likely that a range of processes contribute to explosive activity. These include purely magmatic processes which can contribute to overpressure, and dynamic processes which in large part determine the size and distribution of fracture networks. I also feel that the authors need to do a better job of applying the large range of permeability seen for some individual samples to better comprehend the permeability of the system as a whole. This study has the potential to provide new insight into the lateral and vertical and time-varying permeability distribution of a lava dome system. But to accomplish this, further work and analysis is required.

We're pleased that Reviewer #1 thinks our study "has the potential to provide new insight into the lateral and vertical and time-varying permeability distribution of a lava dome system".

We certainly agree that explosive behaviour can likely be triggered by several, likely cooperative, mechanisms. We have addressed in the submitted manuscript whether hydrothermal alteration can promote explosive volcanic character and, to test this hypothesis, we have considered hydrothermal alteration in isolation. We agree that other factors, such as magma flow rate (a factor that controls the time available for outgassing, cooling, and crystallisation), are important not only for dictating eruptive behaviour, effusive or explosive, but also, as the reviewer points out, for shaping the architecture (e.g., fracture number density) of a resultant dome. However, we consider that the main strength of our study is the demonstration, all else being equal, that hydrothermal alteration alone can promote a switch from effusive to explosive behaviour using our quantitative and multidisciplinary approach. We hope that the reviewer agrees that not only is this of interest, but also that incorporating the multitude of additional parameters known to influence volcanic character would be exceedingly complex (a complexity that would require more assumptions) and would defocus the main aim of the study, which is to understand the role of hydrothermal alteration in promoting explosive behaviour. Although previous authors have qualitatively discussed this, as pointed out by the reviewer, we consider that the role of hydrothermal alteration has been "generally underappreciated" (to quote Reviewer #2). This is in part because no studies have yet used quantitative approaches to tackle this problem, which is what we provide in our manuscript. However, we can now see that we should be clear in the manuscript that hydrothermal alteration is just one factor that can influence eruptive style, so as not to mislead the reader, and that we should also discuss the factors that influence parameters such as the size and number density of fractures within a lava dome. We've now implemented changes to the manuscript in response to this comment, which are outlined below. We also address, in our detailed responses below, the reviewer's comment regarding the range in the laboratory-measured values of permeability.

In the 1990's, Fink and Griffiths showed that for extruding lava domes, both the fracture density and the scale of the fractures are strongly influenced by a balance of the advection time vs. the solidification time. This must be a first-order control upon the distribution of fractures within

an actively growing lava dome. Once a lava dome is emplaced, there must be a balance between magmatic processes such as cooling and crystallization, and hydrothermal processes such as alteration and sealing. Both can lead to pressurization and explosive activity. The trick is understanding this balance in space and time, hence understanding the underlying processes which generate explosive activity. Fractures can be sealed by hydrothermal processes, while new fractures may open from pressure gradients and deeper influences. So there must be a balance here as well. The extent and scale of fracturing are clearly important in this regard.

As outlined above, we consider that assessing the role or importance of hydrothermal alteration in isolation on explosivity as the main strength of our manuscript. However, the reviewer makes some very valid points about the influence of processes such as cooling and crystallisation on not only explosivity, but also on the size and number density of the fractures that exist within a given dome. Understanding the underlying processes that generate explosive activity is, of course, the goal of many volcanologists. However, due to the sheer number of interconnected variables (see, for example, the recent review paper of Cassidy et al. (2018)), which are constrained to different degrees, we hope that the reviewer agrees that insight is perhaps best gained, at this moment, by studying the role certain parameters in isolation. The insight gleaned herein is that, using our multidisciplinary approach, hydrothermal alteration alone can prompt a switch between effusive and explosive volcanism, which we consider to be an important conclusion. However, the reviewer is correct in that other processes are certainly important and, to better inform the reader as to the intention of our contribution, we've added the following text to the manuscript.

“It is understood that volcanic character, effusive versus explosive, depends on a many interconnected parameters (Cassidy et al., 2018). For example, magma flow rate dictates the time available for outgassing, cooling, and crystallisation that, in turn, influence magma viscosity (Cassidy et al., 2018) and the resultant dome morphology, including the number density and morphology of fractures within the dome (Fink and Griffiths, 1998). The goal of this contribution is to quantitatively assess whether hydrothermal alteration alone is sufficient to promote explosive volcanic behaviour.”

A key result from Figure 2 is that the two altered samples show a large range of permeability compared to the unaltered and less altered samples. In fact, a single altered sample appears to have close to four orders of magnitude difference in permeability, i.e., there is no characteristic permeability for these samples. What does this mean? What are the implications for individual samples from a lava dome, and what are the implications for a lava dome as a whole?

The blocks collected from Merapi, a few tens of centimetres in diameter (see photographs in the Supplementary Information), are heterogeneously altered. Typically, the outer edge of an altered block is more altered than the interior. Therefore, multiple 20 mm-diameter cores prepared from different locations within a block will be characterised by different permeabilities, as explained in our manuscript (now changed to improve clarity):

“Our data show that the slightly altered samples (M-SA1 and M-SA2) follow the porosity-permeability trend delineated by the unaltered samples (Figure 1b). However, data from the two highly altered samples (M-HA1 and M-HA2) are characterised by very different porosity-permeability trends (see alteration trajectories indicated by the arrows in Figure 1b). We interpret this variation to be the result of differences in pore-coating,

pore-filling, and microfracture-filling precipitation in the highly altered samples (Figure 2), which greatly decreases permeability, but does not significantly decrease porosity.”

The influence of hydrothermal alteration on matrix permeability was a key ingredient for our permeability modelling and we consider that by measuring core samples prepared from the same block of material that preserve different alteration intensities (see Figure 1d) allows us to quantify the influence of hydrothermal alteration on the matrix permeability of dome rocks. Quantifying the influence of alteration on permeability using differently, but homogeneously, altered blocks would be problematic because other microstructural parameters may differ from block to block, potentially masking the influence as a result of alteration alone.

In terms of implications for individual samples on the dome, these data suggest that the centre of altered blocks is likely more permeable than their exterior. At longer lengthscales, if fluids were required to travel through a block with a low permeability rim, the equivalent permeability of the block would be close to this lowest value of permeability, because the fluid must travel through the different layers in series. In our permeability modelling, which aims to provide the permeability of the lava dome as a whole, we assume that the dome is altered homogeneously. Although this is a simplification, our goal was to avoid the addition of further complexity, which would be difficult to justify, to the numerical model used to model overpressures within and beneath the dome.

Figure 2 also shows some previous data from Kushnir et al. 2016. For slightly altered materials, there is one sample (light green) with comparatively high permeability and one sample (brown) with extremely low permeabilities. How can similarly altered samples have such different permeabilities?

The alteration of our blocks was assessed using microstructural (scanning electron microscope) and mineralogical (X-ray powder diffraction) techniques. Highly altered samples are those that contain a large proportion of alteration phases (gypsum, natroalunite, alunite, quartz, hematite, cristobalite, and unidentifiable amorphous phases (Lines 85-87 in the original submission). Some of these minerals form at the expense of primary minerals. Our assessment of alteration does not take the initial porosity of the material into account. In other words, highly or slightly altered rocks can reflect a range of porosities. (However, if alteration involves the precipitation of alteration phases, it might be unlikely that highly altered rocks will contain very high porosities.) In the case of the slightly altered rocks from the Kushnir et al. paper (light green and dark red), these two blocks contained a similar proportion of alteration phases, and therefore primary phases, but hosted very different porosities (light green = 0.25 and dark red = 0.08). This large difference in porosity is what is controlling the large difference in permeability. Importantly, the permeability of these slightly altered materials is what we would expect from a similarly porous, but unaltered, dome rock (the porosity-permeability trend for the unaltered materials is shown by the grey zone on Figure 1).

Main comments

Line 42: Please explain what you mean by “steam-driven” explosive activity. What is the source of the steam? Meteoric water? Magmatic water? A combination of the two?

This point was also highlighted by Reviewer #2. We now appreciate that it was imprecise to describe these explosions as “steam-driven”, since we consider that these explosions are the result of the build-up of magmatic gases. We have now changed “steam-driven” to “**volatile-driven**” throughout the manuscript.

Line 51: Please provide some further explanation regarding the “near-ubiquity” of alteration at lava domes. At what scales does the alteration occur? How extensive is it? How is it distributed across the surface of a lava dome and within its interior?

The reviewer raises some important questions. Recent electrical tomography at La Soufrière de Guadeloupe, for example, has exposed an extensively altered lava dome, extending 100’s of metres below the surface (Rosas-Carbajal et al., 2016). At Merapi, Byrdina et al. (2017) found, also using electrical tomography, that hydrothermally altered rocks exist below the south and west flanks of the volcano. As for La Soufrière de Guadeloupe, the hydrothermally altered zones extend several 100’s of metres below the surface at Merapi (Byrdina et al., 2017). We now indicate that the alteration can be widespread in our motivation section, and elaborate on the vertical and lateral extent of the alteration later on in our manuscript (see below). We thank the reviewer for prompting this clarification, which has clearly boosted the impact of our study. The improvements to the manuscript are as follows:

“Despite the potential importance of lava dome permeability in regulating volcanic outgassing (Collinson and Neuberg, 2012; Lavallée et al., 2013), and the near-ubiquity of **widespread** alteration at lava domes (Ball et al., 2015; Rosas-Carbajal et al., 2016; Byrdina et al., 2017), no studies thus far have provided upscaled values for the equivalent permeability of hydrothermally altered lava domes to **quantitatively** inform volcanic hazard assessments.”

“**Importantly, recent geophysical imaging at active volcanoes has shown that the vertical and lateral extent of these hydrothermally altered zones can be on the order of a few hundred metres (Rosas-Carbajal et al., 2016; Byrdina et al., 2017).**”

Lines 78-81: I do not see the utility of this figure. The distinctions among the six photos are not shown, nor are they clear to the reader. In the text (lines 105-106), it is stated that the plagioclase crystals are altered with alteration minerals, but the reader cannot clearly see these features.

Although altered plagioclases are indicated on the figure, it is true that these images do not clearly show the alteration of plagioclase crystals. Fracture- and pore-filling alteration is better documented in Figure 2. Since these materials are variably altered, which is a key component of our contribution, we felt it important to (1) provide quantitative mineral contents for our studied materials (we provide XRPD data) and (2) provide a microstructural/microtextural assessment of our studied materials. The goal of these scanning electron microscope images was to inform the reader as to the abundance and size of the pores (easily identifiable in black), the presence of microcracks, and whether there are any textures such as layering. The size and abundance of pores and microcracks are typically of high interest in a laboratory study of permeability, because these are the elements through which the pore fluid passes. To assist the reader, we have now added several new labels on the figure, drawing the reader’s attention to some of the more interesting features. However, upon reflection prompted by the comment of Reviewer #1, we now think that the figure showing the

microstructure, although important, is perhaps better suited for the Supplementary Information. As a result, we have now moved this figure to the Supplementary Information (Supplementary Figure S2).

Lines 192-193: Please justify this assumption that the width and permeability of the Merapi fractures are the same as in your experiments. Why should this necessarily be so?

This is a good question. Since we cannot measure the permeability of the fractures on the dome at Merapi *in situ*, we decided to use the recent approach of Heap and Kennedy (2016), which considers flow in parallel layers. We consider that this permeability measurement, performed on a sample of dome material from Merapi, is more representative than either taking a value from the literature or using the “cubic law” (which assumes straight and parallel fracture surfaces). However, the reviewer is right to question our choice of fracture width. Indeed, it is likely that the fractures on the dome are wider than those generated in the laboratory. Although we cannot access the dome at the moment, due to safety reasons, using recent drone photography we now consider that an average fracture width of 2 cm as appropriate for our modelling. We thank the reviewer for prompting us to reassess our assumptions here. We note that our new permeability modelling, using this new fracture width, does not change the main conclusions of our manuscript. We’ve now changed to manuscript accordingly:

“To upscale our laboratory measurements, we considered a lava dome **with a length of 100 m** that hosts **400 fractures** (a fracture density of 4 m^{-1} is a reasonable estimate for the dome at Merapi volcano; Darmawan et al., 2018a). **We assumed the permeability of these fractures to be the same as determined in our above-described laboratory experiments (i.e. $1.5 \times 10^{-10} \text{ m}^2$) and a fracture width of 2 cm (a reasonable estimate for the fractures within the dome at Merapi volcano).** We considered three scenarios: (1) an unaltered dome with a host rock permeability of $1.0 \times 10^{-13} \text{ m}^2$ in which all fractures are open, (2) a moderately altered dome with a host rock permeability of $1.0 \times 10^{-15} \text{ m}^2$ in which 50% of the fractures are sealed, and (3) a highly altered dome with a host rock permeability of $1.0 \times 10^{-17} \text{ m}^2$ in which 99% of the fractures are sealed. We assumed that a sealed fracture has a permeability of zero. The equivalent permeability of the fractured lava dome for these three scenarios, using Equation 1, is 1.2×10^{-11} , 6.0×10^{-12} , and $1.2 \times 10^{-13} \text{ m}^2$, respectively.”

Lines 200-216: You state in lines 200-205 that progressive sealing to 99% changes the dome permeability by two orders of magnitude. The last 1% of sealing causes another two-order magnitude decline in permeability. In your modeling, however, the large increases in overpressure happen during the first stages of sealing, i.e., at comparatively high permeability. At low permeabilities, the overpressures are essentially constant and do not change. Please explain this apparent paradox.

The reviewer is correct in that permeability is reduced by two orders of magnitude when 99% of the fractures are sealed, and by a further two orders of magnitude when all the fractures are sealed. This is still true when we use the new fracture width of 2 cm for our permeability modelling (see also our answer to the above comment). Although we have also now updated our numerical modelling (in response to a comment of Reviewer #2), we still see large increases in overpressure when the permeability of the dome drops from 10^{-11} m^2 to 10^{-13} m^2 . As before, and as pointed out by the reviewer, further decreases in dome permeability do not result in large increases to the overpressure within and beneath the dome. The reason for this apparent discrepancy is that, to study the

influence of dome permeability in insulation, we kept all the other variables constant in our numerical model. Importantly, the permeability of the country rock (the “edifice”) is fixed. When the permeability of the dome is decreased much below 10^{-13} m^2 , the reason why the overpressure does not continue to increase significantly is because the dome permeability is close to, or lower than, that of the country rock that surrounds the conduit. Higher and lower overpressures within and beneath the dome should be possible if the permeability of the country rock is reduced or increased, respectively. To test this we have now performed new numerical models with different country rock permeabilities (see below). These results show that (1) higher overpressures are indeed capable if the country rock permeability is decreased and (2) the general trend of overpressure increase as a function of dome permeability is similar for the different country rock permeabilities. Importantly, the overpressures generated when using a country rock with a permeability one order of magnitude higher than that used in the manuscript are still capable of fragmenting the majority of dome-forming materials, as now discussed in the revised manuscript (see below).

Rebuttal Figure 1. The influence of country rock permeability on overpressure augmentation as a function of decreasing dome permeability (modelled using a 2D finite element approach in COMSOL Multiphysics V4.3).

“Additional simulations show that a similar pattern of pressure augmentation is seen for domes with different heights (50, 100, and 200 m) and that the magnitude of the overpressure within and beneath the dome depends on the edifice permeability (higher overpressures are possible for lower edifice permeabilities; see Figure S3 in the Supplementary Information).”

“We note that, even if the permeability of the edifice is lowered to 10^{-12} m^2 , the overpressures generated in our highly altered dome scenario are still capable of fragmenting the majority of the rocks and magma within and beneath the dome (see Figure S3 in the Supplementary Information).”

Reviewer #2 (Maarten de Moor)

General comments

This study explores hydrothermal sealing of lava domes as a mechanism leading to explosive volcanic eruptions based on the case of Merapi volcano, Indonesia. The topic is an important one given the hazardous nature of this volcano and these types of eruptions. The approach of the study is innovative and combines petrologic observations, quantitative measurements of connected porosity and permeability, and modelling in order to explain observations of volcanic behavior – an elegant and rigorous combination. The manuscript is well written and the data are robust, though based on a relatively small number of samples (which is understandable considering how difficult and dangerous sample collection is at Merapi – the importance of these samples should not be understated). The arguments are generally reasonable and convincing, i.e. that alteration of lava domes leads to decreased porosity and that this could impede gas flow. Hydrothermal sealing is undoubtedly an important process in some eruptions, and this is a point that is worth stressing as it is generally underappreciated in the literature.

We are pleased to hear that Reviewer #2 considers our study “innovative”, that it offers an “elegant and rigorous combination” of experiments and modelling, that the manuscript is “well written”, and that the data are “robust”. Finally, we agree that hydrothermal sealing is important and that it is underappreciated in the literature.

However, there are number of important issues that need to be addressed. Specifically, more attention needs to be paid to the conditions of the observed alteration and whether this is consistent with sealing of high temperature gas conduits through the dome. Also, the configuration of the model presented in figure 2 does not seem reasonable as the dome is apparently modeled to completely cap the conduit and the surrounding edifice. In reality, lava domes (Merapi is an excellent example) are confined to the conduit area. The authors also argue that the “steam-driven” eruptions at Merapi in 2012, 2014, and 2018 were due to sealing of the dome, however they present no evidence that escape of magmatic gases was inhibited. Therefore, though the manuscript does make a convincing argument that relatively subtle alteration is an important process in decreasing permeability lava domes (an important and provocative finding in itself), it is not fully convincing that this alteration of the dome is in reality responsible for blocking the escape of magmatic gases at Merapi. If these issues can be addressed, I feel that this study would make a good contribution to Nature Communications.

We thank the reviewer for suggestions that helped us greatly improve our manuscript. Our answers to these comments are outlined in detail in our answers below.

Major issues are detailed below, and specific comments are provided in the annotated pdf.

Main comments

1. More details are needed on the conditions of hydrothermal alteration observed in the rock samples. Is this alteration consistent with high temperature alteration by gases in the pathways along which they are travelling, or more consistent with low T alteration/weathering by meteoric fluids with a component of dissolved gas? The former case would be consistent with the proposed model and implications of the manuscript, implying sealing of the principal gas pathways. The latter would imply that the sealing is only affecting the peripheral dome away from the high T gas pathways, which may not be that important for escape of gases (as implied by the results stated on l. 200-202).

The alteration is more consistent with medium- to high-temperature alteration as a result of the circulation of hot, low-pH fluids. Alunite/natroalunite, the most abundant alteration phase, forms as sulphuric acid is produced from the disproportionation and hydration of magmatic SO₂ and/or the oxidation of the H₂S present in rising boiling fluids. These hydrothermal fluids alter the rocks through which they pass in two ways: (1) by reacting with the primary minerals within the rock (e.g., dissolution, replacement) and (2) by precipitating alteration phases within the void space inside the rock (e.g., pores and cracks; see Figure 3 in the manuscript). Therefore, the alteration observed, as the reviewer points out, is consistent with our proposed model and implications. We now state this more clearly in the text:

“Alteration phases, where present, include alunite, natroalunite, quartz, hematite, cristobalite, gypsum, and unidentifiable amorphous phases (Table 1). The most abundant alteration phases—alunite and natroalunite (Table 1)—are stable over a wide range of temperatures (from room temperature to more than 380 °C) and require acidic, oxidising conditions and a fluid with a high sulphate content (Hemley et al., 1969; Stoffregen et al., 2000). We therefore consider that the alteration experienced by these materials was primarily the result of the circulation and cooling of medium- to high-temperature (> 200 °C), acidic (pH < 3) fluids.”

2. Modelling of pore pressures and gas velocity – sensitivity analysis. It is important to conduct sensitivity analysis of the model by varying some of the key variables such as conduit and edifice permeabilities, and provide more information on why the values used were chosen.

Rebuttal Figure 2. Simulation grid from Amy Collinson’s Ph.D thesis showing the sensitivity analysis that informed our model parameter choices.

This is a good point. Since our goal is to better understand the influence of dome alteration on dome permeability, we chose to keep all of the model parameters constant and to vary the dome permeability only. Although most of the simulations presented in the originally submitted manuscript were unique to this study, the model employed was previously described and tested in Collinson and Neuberg (2012). What the reviewer describes, “varying some of the key variables such as conduit and edifice permeabilities”, can be found in Collinson and Neuberg (2012) (see, for example, their Figure 4). The results shown in Collinson and Neuberg (2012) are based on many simulations designed to test the sensitivity of the different parameters (available in Amy Collinson’s Ph.D

thesis; see Rebuttal Figure 2 above). Our goal here was to use this model to better understand how dome permeability influences the development of pore overpressure (a topic only touched upon in Collinson and Neuberg, 2012). We consider the values we chose to be reasonable estimates of the conduit and host rock permeability. However, based on the reviewer's comment here and below, we now provide a model, and suite of new model runs (including model runs that consider a different edifice permeability, see our final answer to the comments of Reviewer #1), that also considers a more realistic dome geometry (see below).

3. Configuration of pore pressure model. The geometry of the model is not realistic. As shown in figure 4, the conduit and edifice are completely capped by a dome, with lateral dimensions of 1300m. In nature, lava domes (at least the ones pertinent to this study) have lateral dimensions of the same order of the dimensions of the conduit, which in this case is narrow (50m?). Details of the model geometry are not given in the text or Supplemental Information. Probably, if the dome has more reasonable dimensions the gas would simply flow around the dome through the more permeable edifice.

This is a very good point, and we have given it much thought. We have now performed a new suite of models to understand how a more realistic dome (one that does not completely cap the edifice; see below) would influence pore pressurisation beneath the dome. Using our more realistic dome shape and lateral extent (see below), we ran models for dome heights of 50, 100, and 200 m. We performed simulations in which we kept the host rock permeability constant (either at 10^{-12} or 10^{-13} m^2), the conduit permeability constant (at 10^{-10} m^2), but varied the dome permeability (from 10^{-11} to 10^{-16} m^2). We performed a total of 36 new simulations. The maximum overpressure for these simulations, as a function of dome permeability is shown in Rebuttal Figure 3 below (a figure that now appears in the Supplementary Information as Figure S3).

Rebuttal Figure 3. Modelling results (modelled using a 2D finite element approach in COMSOL Multiphysics V4.3): a graph of the maximum overpressure in the dome as a function of dome permeability. This figure now appears in the Supplementary Information (Figure S3).

In our revised manuscript, we now show the magnitude and spatial distribution of the overpressure for three representative simulations. We chose to show simulations for the model with an intermediate dome height of 100 m, a host rock permeability of 10^{-13} m^2 , and for dome permeabilities of 10^{-11} to 10^{-13} m^2 . These dome permeabilities are those most relevant for the three scenarios (unaltered, slightly altered, and very altered domes) discussed in our permeability upscaling analysis. The new figure is shown as Rebuttal Figure 4 below.

Rebuttal Figure 4. The magnitude and spatial distribution of overpressure for different dome permeabilities (from 10^{-11} to 10^{-13} m^2). This figure is now included in the manuscript (Figure 3 in the main manuscript).

We have also changed the text in the manuscript to discuss the results of these new models and simulations:

“It is important to assess how a reduction in the equivalent permeability of a dome from 10^{-11} to 10^{-13} m^2 (i.e. unaltered to highly altered) will influence pore pressure augmentation. To do so, we numerically modelled gas loss using a 2D finite element approach in COMSOL Multiphysics V4.3 in which we combined the continuity equation and Darcy’s law, deriving a partial differential equation that was solved for pressure (Collinson and Neuberg, 2012). The model was split into three domains: the conduit, the edifice, and the lava dome (see Figure 3a). To assess the role of dome permeability, we fixed the conduit and edifice permeability at 10^{-10} and 10^{-13} m^2 , respectively, and varied the lava dome permeability from 10^{-11} to 10^{-13} m^2 (the results of additional simulations are provided in Figure S3 in the Supplementary Information). For these three scenarios,

corresponding to the unaltered, slightly altered, and highly altered dome scenarios described above, the maximum overpressure beneath the dome increased from 11.96, to 21.83, and, finally, to 27.14 MPa as dome permeability decreased from 10^{-11} to 10^{-13} m² (Figure 3; tabulated results can be found in Table S1 in the Supplementary Information). Additional simulations show that a similar pattern of pressure augmentation is seen for domes with different heights (50, 100, and 200 m) and that the magnitude of the overpressure within and beneath the dome depends on the edifice permeability (higher overpressures are possible for lower edifice permeabilities; see Figure S3 in the Supplementary Information). We therefore conclude that progressive permeability reduction due to the hydrothermal alteration of a lava dome can significantly increase the pore overpressure within and beneath the dome, leaving the system prone to explosive behaviour.”

We note that changing our model to better capture the geometry of a dome did not alter the main conclusions of our manuscript. Further, we now assess, using the fragmentation threshold, whether these overpressures are sufficient to fragment the material forming the dome (see below).

4. A key argument of the paper is that the “steam-driven” eruptions at Merapi in 2012, 2014, and 2018 are evidence that the hydrothermal sealing process is significant in nature and leads to eruption. However, no evidence is given that magmatic gas flow through the dome was impeded prior to these eruptions (this is a relatively well-monitored volcano – there should be gas data to support the idea). Furthermore, if accumulation of magmatic gases resulted in pore pressure increase and dome failure then these are not steam-driven eruptions but are in fact driven by magmatic gas.

First, we now appreciate that it was imprecise to describe these explosions as “steam-driven”, since we consider that these explosions are the result of the build-up of dominantly magmatic gases. We’ve now changed “steam-driven” to “volatile-driven” throughout the manuscript.

Unfortunately, we do not have access to gas emission data for Merapi during those dates. The monitoring is done by the observatory and not only is there an embargo period on these data, but there is typically a long delay between measurement and publication. However, we now provide some evidence in the form of new time-lapse photography during the May 11 2018 explosion. These photographs suggest that outgassing ceased five days before the explosion, which corroborates the main conclusion of our manuscript, that pore pressure augmentation is linked to explosive behaviour. We have now added an additional figure to the paper (see Rebuttal Figure 5 below) and new text describing the observations:

“A short term sealing process may have led to an increase in pore pressure prior to an explosion in May 2018. Time-lapse photography of the May 11 2018 explosion (Figure 4) highlights that the focussed outgassing at the dome rim (Figure 4a) stopped on May 5 (Figure 4b) and that there was no visible outgassing until the large explosion on May 11 (Figure 4c) (more images are available in Figure S4 in the Supplementary Information). Following the explosion, diffuse outgassing was observed from the dome summit. We interpret the 2018 explosion as a result of the cessation of outgassing caused by hydrothermal sealing, as shown in the accompanying schematic diagrams in Figure 4. Although the appearance of outgassing can depend on environmental factors, such as air temperature and pressure, we note that the presence and absence of outgassing in the

run-up to the May 11 explosion did not depend on time-of-day or changing weather conditions.”

Rebuttal Figure 5. Representative time-lapse photographs of the dome at Merapi volcano before and during the 11 May 2018 explosion. Interpretative schematic diagrams accompany the photographs. More time-lapse photographs are available in the Supplementary Information (Figure S4). This figure appears in the manuscript as Figure 4.

5. More attention is needed on error assessment. Please report errors in Table 2 data and show error bars in Figure 2.

This is a fair point. The experimental error on the porosity measurements can be written as follows:

$$\delta\phi = \phi \times \sqrt{\left(\frac{\delta V_p}{V_p}\right)^2 + \left(\frac{V_b \sqrt{\left(\frac{\delta A}{A}\right)^2 + \left(\frac{\delta l}{l}\right)^2}}{V_b}\right)^2} \quad (R1)$$

where ϕ is the porosity, V_p is the skeletal volume of the sample given by the pycnometer (precision = 0.00005 cm³), A is the cross-sectional area of the sample (precision on radius = 0.0005 cm), l is the length of the sample (precision on length = 0.0005 cm), and V_b is the bulk volume of the sample (precision on length and radius = 0.0005 cm). The experimental error of our porosity measurements is <<1%. The experimental error on the permeability measurements can be written as follows:

$$k = k \times \sqrt{\left(\frac{\delta v}{v}\right)^2 + \left(\frac{\delta l}{l}\right)^2 + \left(\frac{\delta A}{A}\right)^2 + \left(\frac{\delta P_d}{P_d}\right)^2 + \left(\frac{\delta P_m}{P_m}\right)^2} \quad (R2)$$

where k is the permeability, v is the volumetric flow rate (precision on flow rate = 8.33×10^{-10} m³/s), l is the length of the sample (precision on length = 0.0005 cm), A is the cross-sectional area of the sample (precision on radius = 0.0005 cm), and P_d and P_m are the downstream and mean pore fluid pressure, respectively (precision on the pressure gauge = 5 Pa). The experimental error on our permeability measurements is therefore <1%. Since these errors are within the symbol size of the symbols on Figure 1, there is no need to add error bars. We have now indicated in the caption of Figure 1 and Table 2 that the error on these measurements is <1%.

Comments from the annotated manuscript

Line 30: Why the inverted commas?

We've now deleted the inverted commas.

Lines 32-33: Should propose how to monitor hydrothermal alteration.

We've now added a suggested method: "...is monitored at active dome-forming volcanoes using geophysical techniques (e.g., electrical or muon tomography) and/or gas monitoring and incorporated into real-time hazard assessments."

Line 44: Galeras.

Corrected.

Lines 66-69: When were the blocks extruded?

The blocks were extruded in 1902. We now mention this in the text:

"These blocks, extruded in 1902, were selected as representative of the various degrees of visually discernible alteration present."

Line 71: samples.

In the manuscript we refer to large blocks collected from the volcano as "blocks" and the cores prepared from them as "samples". To avoid any confusion, we'd prefer to keep "blocks" here, rather than change it to "samples".

Line 72: Please include photo of this block in Supp Info.

We've now added pictures of the 2006 block to the Supplementary Information (Figure S1).

Lines 75-76: Please explain what you mean and the importance of this. It is not clear if the samples were in place when sampled and if oriented samples were collected.

The blocks collected were not orientated. However, since volcanic rock blocks can be anisotropic in terms of their texture, porosity, and therefore permeability, we decided to core all of the 20 mm-diameter samples in the same orientation. We have now reworded this sentence to improve clarity accordingly:

“We measured the connected porosity and permeability of between ten and eleven cylindrical core samples extracted from each of the five main blocks (cores from the same block were all cored in the same orientation), as well as five core samples prepared from the 2006 block (57 core samples in total) (see Methods).”

Lines 86-87: What are T, pH, and redox conditions leading to this assemblage?

We now provide information on the conditions required for this alteration assemblage:

“The most abundant alteration phases—alunite and natroalunite (Table 1)—are stable over a wide range of temperatures (from room temperature to more than 380 °C) and require acidic, oxidising conditions and a fluid with a high sulphate content (Hemley et al., 1969; Stoffregen et al., 2000). We therefore consider that the alteration experienced by these materials was primarily the result of the circulation and cooling of medium- to high-temperature (> 200 °C), acidic (pH < 3) fluids.”

Line 103: few.

Changed as suggested.

Lines 105-106: Please describe this alteration in more detail. What secondary minerals are present?

These fractures are often sealed with alunite/natroalunite. We now specify this in the text: “...and often contain fractures and pores that are sealed with alteration minerals, often alunite or natroalunite.”

To add further clarity, we now also identify the secondary/alteration phases (with asterisks) in Table 1.

Line 112: By aqueous fluids or by gas-rock interaction?

We now provide more information as to the conditions required for this alteration assemblage. See our answer to an above comment.

Lines 113-116: Also see recent work by Rodriguez and van Bergen: Rodríguez, A., and M. J. van Bergen (2017), Superficial alteration mineralogy in active volcanic systems: An example of Poás volcano, Costa Rica, Journal of Volcanology and Geothermal Research, 346, 54-80. Also see our new paper (in press) proposing hydrothermal sealing as the primary mechanism for the

explosive dome-destroying eruption at Poas in 2017:
<https://agupubs.onlinelibrary.wiley.com/doi/abs/10.1029/2018GL080301>.

We thank the reviewer for bringing these interesting and relevant papers to our attention. We now cite them here (and elsewhere in the manuscript):

“...Citlaltépetl (Zimbelman et al., 2005) (Mexico), Vulcano (Boyce et al., 2007) (Italy), Whakaari volcano (Hedenquist et al., 1993; Heap et al., 2017) (New Zealand), and Poás volcano (Rodríguez and van Bergen, 2017; de Moor et al., 2019).”

“For example, recent gas monitoring (SO₂/CO₂ and SO₂ fluxes) at Poás volcano (Costa Rica) led to models suggesting that hydrothermal sealing may have been the cause of the explosive phreatomagmatic eruption in 2017 (de Moor et al., 2019).”

Line 126: Should state here the observed difference between altered and unaltered samples.

We agree. We have now moved a couple of paragraphs from the discussion section to the results section that describe these differences. On reflection, we think these paragraphs are better suited in the results section.

Figure 2: Need error bars on the data points.

As explained above, the measurement error is within the size of the symbol. However, we now also report the experimental errors (<1%) in the captions of Figure 1 and Table 2.

Figure 2: c and d are probably not necessary. d can be integrated into a.

We consider it informative to show photographs of the unaltered samples from Kushnir et al. (2016) to highlight the visible differences between the altered and unaltered samples. As a result, we'd prefer, in this instance, to keep these photographs as part of Figure 1.

Lines 153-154: Should state that fractures are fluid pathways and therefore also alteration and deposition sites.

Agreed. We now make this point a little clearer:

“This is because, although microfractures provide important flow paths in volcanic rocks (Farquharson et al., 2015), they represent only a small volume of the porosity within the rock. Therefore, when these microfractures are sealed or partially sealed (Figure 2b and 2e) as minerals precipitate from the circulating hydrothermal fluids, a small decrease in sample porosity can result in a considerable permeability reduction.”

Line 190: Ten metres long in what dimension? The model in figure 4 depicts a dome with thickness of 300m that is completely capping both the conduit and edifice, which is not a realistic configuration.

10 m in length. Although it doesn't change the calculation, we've decided to consider a length of 100 m, to avoid readers thinking that we are only considering a small portion of the dome. We've changed the text as follows:

“To upscale our laboratory measurements, we considered a lava dome with a length of 100 m that hosts 400 fractures (a fracture density of 4 m^{-1} is a reasonable estimate for the dome at Merapi volcano; Darmawan et al., 2018a).”

We agree that a 300 m-thick dome that caps the entire edifice may not represent a realistic configuration. As outlined above, we now provide new modelling that uses a more realistic geometry.

Lines 210-211: A better explanation of the model configuration is needed. Why is the dome completely capping the edifice and conduit? Domes are typically similar in dimension to the conduit.

This question is the same as comment three in the reviewer’s main comments. We have updated our model configuration such that the dome geometry is more realistic (see above).

Line 212: Need to explain where these values come from and why they were chosen. How sensitive is the model to varying these parameters?

This question is the same as comment two in the reviewer’s main comments. Our response can be found above.

Line 216: Does not make sense here as you have not explained that there six scenarios were modeled.

We appreciate that this was not explained clearly. To improve clarity, we now only show simulations for the three scenarios discussed in our permeability upscaling analysis. Further, we now better introduce why these model runs were performed:

“To assess the role of dome permeability, we fixed the conduit and edifice permeability at 10^{-10} and 10^{-13} m^2 , respectively, and varied the lava dome permeability from 10^{-11} to 10^{-13} m^2 (the results of additional simulations are provided in Figure S3 in the Supplementary Information). For these three scenarios, corresponding to the unaltered, slightly altered, and highly altered dome scenarios described above, the maximum overpressure beneath the dome increased from 11.96, to 21.83, and, finally, to 27.14 MPa as dome permeability decreased from 10^{-11} to 10^{-13} m^2 (Figure 3; tabulated results can be found in Table S1 in the Supplementary Information).”

Line 219: This would require mechanical failure of the dome. At what point would that happen? Can you comment on the effects of alteration on the mechanical strength of the dome?

These are very interesting points. We’ve now added an additional paragraph (also in response to Reviewer #3) that discusses how these overpressures could fragment the material within and/or beneath the dome, and how alteration could further weaken the dome and promote fragmentation:

“In a next step we assess whether the overpressures predicted by our modelling (Figure 3) are capable of fragmenting rock and magma. The fragmentation criterion, derived from the stress distribution surrounding isolated spherical pores (Koyaguchi et al., 2008), has been shown to well describe the available experimental data for the fragmentation threshold, P_{th} , for volcanic rocks and magmas:

$$P_{th} = \frac{2S(1 - \phi)}{3\phi\sqrt{\phi^{-1/3} - 1}}, \quad (3)$$

where S and ϕ are the effective tensile strength and porosity of the material, respectively. Using a value of S that well describes experimental data for andesites from Volcán de Colima (Mexico; Lavallée et al., 2019), Equation (3) suggests that the maximum overpressure modelled beneath an unaltered dome characterised by a permeability of 10^{-11} m^2 (11.96 MPa; Figure 3b) is capable of fragmenting material with a porosity of ~ 0.16 . An increase in overpressure to 27.14 MPa (i.e. the highly altered dome scenario, Figure 3d) allows for the fragmentation of material with a porosity as low as ~ 0.05 . Porosity values for the rock samples measured herein vary from 0.08 to 0.28 (Table 2), and laboratory porosity values for historical dome samples vary significantly, from ~ 0.01 up to ~ 0.5 (Le Pennec et al., 2001; Kushnir et al., 2016). Electromagnetic tomography at Merapi has yielded porosity estimates of 0.05-0.1 (Commer et al., 2005). An increase in overpressure under the dome from 11.96 to 27.14 MPa is therefore sufficient to fragment the vast majority of the rocks and magma within and beneath the dome at Merapi volcano. Further, if hydrothermal alteration also reduces the effective tensile strength of the dome materials (Pola et al., 2012; Wyering et al., 2014; Heap et al., 2015; Farquharson et al., 2019), the fragmentation threshold of a rock with a given porosity will be lowered. We note that, even if the permeability of the edifice is lowered to 10^{-12} m^2 , the overpressures generated in our highly altered dome scenario are still capable of fragmenting the majority of the rocks and magma within and beneath the dome (see Figure S3 in the Supplementary Information)."

Figure 4: If my interpretation of the figure is correct, the model is assuming a 1300 m wide dome! This is not realistic. The authors need to consider a more reasonable dome configuration with width similar to that of the conduit. In this case, would the gas not simply flow around the dome? Also, the numbers are very small in this figure and very hard to read at reasonable zoom. X axis also needs to be labeled.

Agreed. As explained above, we have now updated our model configuration such that the dome geometry is more realistic. Further, we have made the numbers more legible and have added a scale bar.

Lines 229-231: Not necessarily. As long as there is degassing of acid gas species there will be acid alteration of the surface. Please compare the mineral assemblage of the surface alteration to that observed in the dome rocks.

Fair point. Unfortunately, we do not have the mineral assemblage of the surface alteration. Access to the dome during this period was not possible. We think it's likely, however, that the increase in alteration observed on the surface of the dome also corresponds to an increase in alteration within the dome. We've now reworded this sentence so as to not be so matter-of-fact:

"A final consideration is the time required to produce widespread alteration of a lava dome. At Merapi volcano, for example, sequential images of the lava dome (taken using a drone) show that secondary mineral deposition at the surface, which **we consider here to be also** associated with significant alteration at depth, can develop in less than three years (Darmawan et al., 2018b)."

Lines 231-234: In this case there should have been an observed decrease in the SO₂ flux prior to the explosions. Did this occur? These data should be available. Also, what is the evidence that these were "steam-driven" eruptions? If the implication is that these were not driven by magmatic gas, then this does not seem consistent with your proposed mechanism for the eruptions, which is that magmatic gas could not escape through the impermeable dome.

Unfortunately, as outlined above, we do not have access to the gas monitoring data. We have also now changed instances of "steam-driven" to "volatile-driven", as also explained above. In the absence of gas emission data, we now provide evidence, in the form of time-lapse photographs, for hydrothermal sealing promoting explosive behaviour (see above).

Lines 246-249: Is this the case for your rocks? Acid dissolution of primary phases increases permeability and porosity and is associated with weakening (Farquharson et al., 2019, JGR). In the case of Merapi porosity and permeability decrease with alteration, conceivably associated with strengthening.

This is a very good question. However, although we observe a porosity decrease due to alteration, the alteration phases, such as alunite, are weak minerals (compared to the primary minerals). A net decrease in sample porosity also does not rule out local increases in porosity (creating weak zones that can act as nuclei for macroscopic failure), or more widespread porosity increase in, for example, the groundmass (through the formation of micropores). In the absence of strength data for these materials, we rely on the conclusions of previous studies on the influence of alteration on the strength of volcanic rocks.

Lines 249-251: Also need to mention gas monitoring as this provides real-time data on hydrothermal sealing processes. See de Moor et al. (in press).

Agreed. We now mention gas monitoring in the text and reference de Moor et al. (2019):

"On the basis of our findings, mapping the extent and evolution of hydrothermal alteration at active lava domes using geophysical methods such as electrical (Rosas-Carbajal et al., 2016; Byrdina et al., 2017; Ghorbani et al., 2018; Ahmed et al., 2018) and muon tomography (Lesparre et al., 2012; Rosas-Carbajal et al., 2017; Le Gonidec et al., 2019), spectroscopic methods such as visible and infrared spectroscopy (Crowley and Zimelman, 1997; John et al., 2008), and/or gas monitoring (de Moor et al., 2019) emerge as an important tools to help anticipate dome explosions at otherwise unpredictable dome-forming volcanoes."

Reviewer #3 (Cristian Montanaro)

General comments

The submitted manuscript presents an experimental study, integrated by numerical modelling, on the effect of hydrothermal alteration on lava domes as one process controlling their explosive behavior. The authors performed mineralogical and petrophysical characterization of series of rock samples from Merapi volcano, representative of basaltic-andesitic/andesitic volcano-type of rocks, and differently altered by acid-sulfate alteration. Laboratory measurements indicate that (moderate to highly) altered dome rocks from Merapi, not only are less permeable than their porosity would suggest, but their permeability also varies by up to four orders of magnitude ($\sim 10^{-13}$ to $\sim 10^{-17} \text{m}^2$) within a narrow porosity range (~ 0.15 to ~ 0.25). The permeability of artificially fractured specimens was also estimated to be in the order of $\sim 10^{-10} \text{m}^2$. To upscale the laboratory measurements authors simulated, by means of a numerical modeling, a ten-meter-long dome with unaltered to moderately and altered rocks, having fractures differently sealed (from complete open system to 99% sealed fractures). Numerical results show that intense alteration can reduce the rocks/fractures permeability of a dome by two orders of magnitude switching the outgassing regime from “open” to “partially closed”, thus favoring pore pressurization within and beneath the dome, and priming the system for explosive behaviour. Results are significant, and shed a quantitative light on the very important effect of the hydrothermal alteration as a priming factor for volcanic explosions at volcanic domes, therefore the study is definitely worthy of publication. I recommend publication after that the following comments (minor) have been suitably addressed.

We're pleased that the reviewer considers our results “significant” and our study “definitely worthy of publication”.

Main comments

Starting from the manuscript Title... The authors do only explore how acid-sulfate alteration affects dome rocks from the basaltic-andesitic and andesitic types. It is true that those are the most widely distributed (and erupting) once, though it would be worthy to discuss how similar alteration processes may affect different dome rock compositions (e.g. dacite, rhyodacite or rhyolite?) to put the study's results in a broader context. If not I think a title like “Hydrothermal alteration of a andesitic dome leads to explosive volcanic behavior” would be more appropriate.

Although we do give examples of domes with different compositions (e.g., the dacitic domes of the Cascade volcanoes, USA) that are characterised by similar acid-sulphate hydrothermal alteration, the reviewer is correct in that it is perhaps overreaching to assume that domes of all compositions would behave in a similar manner (our results may not be directly applicable to the rhyolitic domes of Chaitén and Cordon Caulle (Chile), for example). We have therefore, as suggested by the reviewer, slightly modified the title to incorporate the word “andesitic”:

“Hydrothermal alteration of **andesitic lava domes leads to explosive volcanic behaviour”**

In the Results, line 123-126, the sentence is very long and not clear in some points. I suggest breaking it in two, and specify in line 124 that you talk about the trend observed for unaltered rocks. “respectively. In particular it can be observed how the permeability of unaltered dome samples”

We agree. We have now split the long sentence into two sentences, and we now explicitly mention that we're talking about the unaltered rocks:

“These data show that the porosity and permeability of unaltered dome rock from Merapi can vary from ~0.08 to ~0.28 and from $\sim 2 \times 10^{-17}$ to $\sim 1 \times 10^{-11}$ m², respectively (Figure 1b). We also note that the permeability of the unaltered dome rock increases as connected porosity is increased (indicated by the grey zone in Figure 2b), in agreement with many published studies for unaltered andesites and basaltic-andesites worldwide (Mueller et al., 2005; Melnik and Sparks, 2002; Bernard et al., 2007; Heap et al., 2014; Farquharson et al., 2015; Heap and Kennedy, 2016).”

Additionally, the deviation in permeability/porosity of altered rocks, in respect to the general trend, should be discussed briefly to highlight the decrease of permeability due to pore-filling minerals, and likely to the formation of complex connected pore network.

We agree. We have now moved some text from the discussion section that we now consider is better suited in the results section:

“The porosities and permeabilities of the slightly altered samples (M-SA1, M-SA2), M-2006 (cristobalite alteration), and the cristobalite-bearing samples of Kushnir et al. (2016) follow the trend of the unaltered rocks (indicated by the grey zone in Figure 1b). However, not only are the core samples from the highly altered blocks (M-HA1 and M-HA2) less permeable than their porosity would suggest, but their permeability also varies by up to four orders of magnitude, despite their narrow porosity range. For example, although the difference in porosity between samples M-HA1-10 and M-HA1-2 (samples cored from the same block) is only 0.03, their permeabilities are 2.1×10^{-13} and 4.9×10^{-17} m², respectively (Figure 2b; Table 2).”

In Line 143, it is stated that the slightly altered material has permeability/porosity following the main trend of unaltered rocks, however, they seems already to show a slight shift down in permeability (likely due to the pore-filling effect), for samples having similar porosity range. In case, I think this should be briefly discussed.

Whether there is a slight shift in permeability in the slightly altered rocks is very difficult to state with confidence, especially when one considers the expected natural variability in the permeability of volcanic rocks measured in the laboratory. Since these materials do not significantly differ in permeability with respect to the unaltered materials, for a given porosity, we'd prefer to conclude as such:

“The porosities and permeabilities of the slightly altered samples (M-SA1, M-SA2), M-2006 (cristobalite alteration), and the cristobalite-bearing samples of Kushnir et al. (2016) follow the trend of the unaltered rocks (indicated by the grey zone in Figure 1b).”

In Line 144, please state that you are talking about “Highly altered material (M-HA1 and M-HA2)”

Good point. We have now improved the clarity of this sentence: “However, not only are the core samples from the highly altered blocks (M-HA1 and M-HA2) less permeable than...”

In Lines 154-156, here should also be discussed how the formation of very complex, and

tortuous, pore network may influence/reduce samples permeability having same/similar porosity (e.g. Colombier, M., Wadsworth, F.B., Gurioli, L., Scheu, B., Kueppers, U., Di Muro, A., Dingwell, D.B., 2017. The evolution of pore connectivity in volcanic rocks. *Earth Planet. Sci. Lett.* 462, 99–109; Kushnir, A.R., Martel, C., Bourdier, J.L., Heap, M.J., Reuschlé, T., Erdmann, S. Cholik, N., 2016. Probing permeability and microstructure: unravelling the role of a low permeability dome on the explosivity of Merapi (Indonesia). *J. Volcanol. Geotherm. Res.* 316, 56–71.)

Agreed. We now include an additional sentence to emphasise this point:

“Indeed, volcanic rock samples with similar porosities can be characterised by very different permeabilities, a function of the connectivity of their void space (Colombier et al., 2017).”

An interesting point is that alteration leads to the formation of micro-pores (tens of micron), which can have quite an important role in case of steam-driven explosive events where liquid water is filling dome rocks (likely to occur, especially in rainy regions). Indeed, water explosivity may be enhanced by the micro pores structure, as theoretically postulated (Thiéry, R., Looock, S., & Mercury, L. (2010). Explosive properties of superheated aqueous solutions in volcanic and hydrothermal systems. In *Metastable Systems under Pressure* (pp. 293–310), and qualitatively observed in experiments (Montanaro, C., Scheu, B., Mayer, K., Orsi, G., Moretti, R., Isaia, R., & Dingwell, D. B. (2016). Experimental investigations on the explosivity of steam-driven eruptions: a case study of Solfatara volcano (Campi Flegrei). *Journal of Geophysical Research: Solid Earth*, 121(11), 7996–8014. <https://doi.org/10.1002/2016JB013273>) Maybe this point can be briefly discussed here, and recalled later when discussing potential effect of explosive events in destabilizing domes: for e.g. more energetic explosive event may favor larger removal of dome material.

These rocks do contain altered plagioclase crystals that appear to contain abundant microporosity (see Figure S2). However, we highlight that it’s unclear whether such alteration has resulted in the formation of micropores within the groundmass. The complex microstructure presented by the tuffs (i.e. lots of micropores) studied in Montanaro et al. (2016) results in a somewhat unexpected low sample permeability, given their high porosity, which greatly influences at what pressure they fragment during decompression experiments. Although we agree that the sealing of microcracks in our samples will force fluids to travel through more tortuous pathways within the rock, we’re unsure whether isolated plagioclase phenocrysts containing micropores will play a significant role. For this reason, we’d prefer to avoid any discussion of the role of microporosity here, which would necessarily be highly speculative.

Lines 213-219: Once a certain overpressure is reached, rock porosity play a further important role in i) storage the gas potential energy, as well as in ii) determining the energy/pressure threshold needed for starting the fragmentation (e.g. Scheu, B., Kueppers, U., Mueller, S., Spieler, O., & Dingwell, D. B. (2008). Experimental volcanology on eruptive products of Unzen volcano. *Journal of Volcanology and Geothermal Research*, 175(1–2), 110–119. <https://doi.org/10.1016/j.jvolgeores.2008.03.023>; Richard, D., Scheu, B., Mueller, S. P., Spieler, O., & Dingwell, D. B. (2013). Outgassing: Influence on speed of magma fragmentation. *Journal of Geophysical Research: Solid Earth*, 118(3), 862–877. <https://doi.org/10.1002/jgrb.50080>). Though the model does not account for this parameter, I think is interesting to discuss (later for e.g. between lines 228-243) the effect of rocks porosity considering those measured from Merapi's samples. For e.g. for dacitic dome rocks having porosities in the range 0.15 to 0.25 an overpressure >10 and <20 MPa is needed

for fragmentation, though rock strength weakening due to alteration may reduce the fragmentation threshold (see next comment). Moreover, and considering that permeability determines whether the expanding fluid may either fragment the surrounding rocks (permeability $<\sim 10^{-12} \text{m}^2$) or escape from it via effective outgassing (permeability $>\sim 10^{-12} \text{m}^2$; Richard et al., 2013), at Merapi only highly altered rocks (in the 0.15 to 0.25 porosity range) have permeability lower enough ($<10^{-13} \text{m}^2$) to favor fragmentation. I think these arguments may further strengthen the results found in this study.

These are great points. We have now added an additional paragraph that discusses how these overpressures could fragment the material within and/or beneath the dome, and how alteration could further weaken the dome and promote fragmentation. We thank the reviewer for prompting these changes which, in our opinion, have significantly strengthened our contribution. We have now added the following section to the manuscript:

“In a next step we assess whether the overpressures predicted by our modelling (Figure 3) are capable of fragmenting rock and magma. The fragmentation criterion, derived from the stress distribution surrounding isolated spherical pores (Koyaguchi et al., 2008), has been shown to well describe the available experimental data for the fragmentation threshold, P_{th} , for volcanic rocks and magmas:

$$P_{th} = \frac{2S(1 - \phi)}{3\phi\sqrt{\phi^{-1/3} - 1}}, \quad (3)$$

where S and ϕ are the effective tensile strength and porosity of the material, respectively. Using a value of S that well describes experimental data for andesites from Volcán de Colima (Mexico; Lavallée et al., 2019), Equation (3) suggests that the maximum overpressure modelled beneath an unaltered dome characterised by a permeability of 10^{-11}m^2 (11.96 MPa; Figure 3b) is capable of fragmenting material with a porosity of ~ 0.16 . An increase in overpressure to 27.14 MPa (i.e. the highly altered dome scenario, Figure 3d) allows for the fragmentation of material with a porosity as low as ~ 0.05 . Porosity values for the rock samples measured herein vary from 0.08 to 0.28 (Table 2), and laboratory porosity values for historical dome samples vary significantly, from ~ 0.01 up to ~ 0.5 (Le Pennec et al., 2001; Kushnir et al., 2016). Electromagnetic tomography at Merapi has yielded porosity estimates of 0.05-0.1 (Commer et al., 2005). An increase in overpressure under the dome from 11.96 to 27.14 MPa is therefore sufficient to fragment the vast majority of the rocks and magma within and beneath the dome at Merapi volcano. Further, if hydrothermal alteration also reduces the effective tensile strength of the dome materials (Pola et al., 2012; Wyering et al., 2014; Heap et al., 2015; Farquharson et al., 2019), the fragmentation threshold of a rock with a given porosity will be lowered. We note that, even if the permeability of the edifice is lowered to 10^{-12}m^2 , the overpressures generated in our highly altered dome scenario are still capable of fragmenting the majority of the rocks and magma within and beneath the dome (see Figure S3 in the Supplementary Information).”

Lines 246-249: I think is worthy to add references about the effect of acid-sulfate alteration in decreasing rocks strength as already observed in other volcanic terrains (e.g. Solfatara in Campi Flegrei, Italy – e.g. Mayer, K., Scheu, B., Montanaro, C., Yilmaz, T. I., Isaia, R., Aßbichler, D., & Dingwell, D. B. (2016). Hydrothermal alteration of surficial rocks at Solfatara (Campi Flegrei): Petrophysical properties and implications for phreatic eruption processes. Journal of

Volcanology and Geothermal Research, 320, 128–143. <https://doi.org/10.1016/j.jvolgeores.2016.04.020>) or other andesitic domes (e.g. Santiaguito dome complex in Guatemala, e.g. Ball, J. L., Calder, E. S., Hubbard, B. E., & Bernstein, M. L. (2013). An assessment of hydrothermal alteration in the Santiaguito lava dome complex, Guatemala: Implications for dome collapse hazards. *Bulletin of Volcanology*, 75(1), 1–18. <https://doi.org/10.1007/s00445-012-0676-z>), since the jointed effect of permeability and strength reduction may i) lower the rock fragmentation threshold (thus reducing overpressures needed for explosive failures) and favor (minor to large) flank insatiability.

Agreed. We've now added these references:

“We further note that hydrothermal alteration typically weakens volcanic rock (Pola et al., 2012; Ball et al., 2013; Wyring et al., 2014; Heap et al., 2015; Mayer et al., 2016) and that such weakening could reduce the stability of the dome...”

Minor corrections

Line 26: “rock samples”

Modified to: “New laboratory data show that acid-sulphate alteration, common to the domes of active volcanoes worldwide, can reduce the permeability of dome rock on the sample lengthscale by up to four orders of magnitude.”

Line 71: “supplemented”

Changed as suggested.

Line 85-86: “Alteration phases present in the samples included”

The alteration phases are still present in the samples. We'd prefer to keep with the present tense in this instance.

Lines 112-117: “domes, and craters of many...” (otherwise Vulcano and Whakaari volcano should be cut from the list)

Good point. Changed as suggested.

Lines 120: “alongside with”

We would prefer, in this instance, to keep “alongside”.

Line 150: “manifested as”

During our reorganisation of this section, “manifest” was omitted.

Line 177: “from the unaltered block”

Modified as follows: “...we prepared two additional samples from unaltered block M-U (25 mm in diameter and 25 mm in length).”

Line 182: “remeasured” ... by using a permeameter? In case just specify and add that you are

measuring permeability of fractures and connected pore, and after the permeability due to fractures can be estimated using the formula 2.

The permeability was measured using the same permeameter and using the same procedure. We now provide this information, and further clarification on the nature of the measured sample, in the text:

“The permeability of the now-fractured samples (i.e. the permeability of a sample containing two “intact” portions separated by a fracture) was then remeasured using the same laboratory procedure (see Methods section).”

Line 188: “equivalent”(?)

The equivalent permeability, in our opinion, would describe, for example, the permeability of a sample containing a fracture. Here we discuss the average permeability of the resultant fracture.

Figure 2: Black circles are not in the plot. Are they missing? Covered by other data symbol? Or the color of some of the data symbols is wrong? Please check.

There were no black symbols. The symbols for M-U are dark grey (although we appreciate that it’s not so clear). To avoid any future confusion, we have now changed the colour of these symbols to black.

Figure 4: can the $\times 10^7$ in the OP scale bar be removed?

Figure 4 (now Figure 3 in the revised manuscript) has now completely changed thanks to the comments of Reviewer #2.

Comments from the annotated manuscript

The comments on the annotated manuscript are the same as those outlined above. We therefore consider that these comments have been addressed (see above).

Reviewers' comments:

Reviewer #1 (Remarks to the Author):

General remarks

Strong points: The trends shown on Figure 1 are very interesting and thought-provoking. The finding that altered lava dome rocks follow a porosity-permeability path distinct from unaltered dome rocks is novel and significant. The lack of correlation between porosity and permeability for these altered rocks is also a significant result.

Weak points: The suggestion that permeability changes drastically for the last 1% of sealing is not correct. The conduit and edifice permeability are not properly evaluated. There are no SO₂ data presented to support the hypothesis of sealing.

Detailed comments

Line 25: Clarify what you mean by "sample lengthscale" and place some quantitative values on this rather poorly defined term.

Line 28: You say nothing about mechanisms in the abstract. I suggest adding something such as "by two orders of magnitude from hydrothermal mineral precipitation in fractures and consequent sealing" or some such thing.

Line 35: Replace "and/or" with "and".

Line 64: "upscaled value"...this colloquial term should be rewritten.

Line 90: In the paper you discuss short timescales of sealing, potential sealing events at Merapi during 2010-2019, etc. (e.g., line 308). That is fine. But here, unless there is a typo, you state that

you collected samples from dome material extruded in 1902, more than 100 years ago. Please justify this sampling and state why you did not collect the 2010-2019 dome material.

Line 110: Block M-U is the least altered, is highly microfractured, yet according to Figure 1 appears to have some of the lowest permeabilities of all samples studied. So what is this telling us about microfractures influencing permeability?

Figure 1: The reader cannot easily distinguish among M-U, M-SA2, and M-HA2 because the symbols look so similar. Please fix this.

Line 198: As you state, there are at least 4 orders of magnitude here, and this is a minimum value. It could be more.

Lines 247-250. Sorry, this makes no sense. You are saying that there is a four order of magnitude change in permeability from 99% sealing to 100% sealing. By 99% sealing the system is essentially closed already, so a 1% increment should not change things.

Line 258: You do not justify your choices of conduit and edifice permeability, which have a huge difference. I could easily argue that the values should be reversed. The edifice should be highly permeable, especially at Merapi which is built of pyroclastic material. The conduit is a plug of magma which may have extremely low permeability.

Figure 3: The figure is labelled "host rock" but what you mean here is "edifice" do you not? Otherwise the reader is confused by "host rock" referring to dome material on lines 239-243, and "host rock" referring to the edifice in Figure 3.

Lines 315-330: The best monitoring evidence for sealing are changes in SO₂ flux as a function of time. Merapi is a well monitored volcano for which SO₂ data are certainly available.

Line 360: Replace "and/or" with "and".

Reviewer #2 (Remarks to the Author):

The authors did a nice job of addressing the comments in the 1st round of reviews. The revised study is now more convincing. This is a thought-provoking manuscript that I am happy to endorse for publication.

Reviewer #3 (Remarks to the Author):

Dear Dr. Heap and colleagues,

The submitted and revised manuscript "Hydrothermal alteration of andesitic lava domes leads to explosive volcanic behaviour" include all changes in response to my comments.

Regards the rebuttals, the authors decided not to modified some of the points I was making, but their justification/argumentation were valid and well argued, thus I am fine with the authors responses.

Some of the manuscript paragraphs/section I was commenting, now appear further improved also thanks to the comments of the other referees.

Overall, I do not have further comments/corrections on the submitted version which, in my opinion, is to be totally accepted for publication.

Only a minor thing, please change my name in the acknowledgments to Cristian without "h".

All the best,

Cristian Montanaro

Reviewer #1

General remarks

Strong points: The trends shown on Figure 1 are very interesting and thought-provoking. The finding that altered lava dome rocks follow a porosity-permeability path distinct from unaltered dome rocks is novel and significant. The lack of correlation between porosity and permeability for these altered rocks is also a significant result.

We are pleased by these endorsements from Reviewer #1. We are in agreement that our results are “novel and significant”.

Weak points: The suggestion that permeability changes drastically for the last 1% of sealing is not correct. The conduit and edifice permeability are not properly evaluated. There are no SO₂ data presented to support the hypothesis of sealing.

We thank the reviewer for highlighting these points. Our responses to these comments are listed below.

Detailed comments

Line 25: Clarify what you mean by “sample lengthscale” and place some quantitative values on this rather poorly defined term.

We have now clarified “sample lengthscale” in the abstract (see below). We have also added quantitative values here, and when we discuss the permeability of the dome:

“...can reduce the permeability of dome rock on the lengthscale of a laboratory sample (i.e. length of 40 mm) by up to four orders of magnitude. These data are used to provide estimates for the upscaled permeability of fractured, altered lava domes (i.e. several tens of metres).”

Line 28: You say nothing about mechanisms in the abstract. I suggest adding something such as “by two orders of magnitude from hydrothermal mineral precipitation in fractures and consequent sealing” or some such thing.

The reviewer raises a good point. We have now included an additional sentence in the abstract that discusses the cause of the permeability reduction:

“Permeability reduction is the result of pore- and microfracture-filling precipitation of alteration minerals, particularly alunite.”

Line 35: Replace “and/or” with “and”.

Changed as suggested.

Line 64: “upscaled value”...this colloquial term should be rewritten.

Since this sentence already describes the equivalent permeability of a lava dome, the term “upscaled” is actually not needed and has therefore been removed to avoid any possible confusion.

Line 90: In the paper you discuss short timescales of sealing, potential sealing events at Merapi during 2010-2019, etc. (e.g., line 308). That is fine. But here, unless there is a typo, you state that you collected samples from dome material extruded in 1902, more than 100 years ago. Please justify this sampling and state why you did not collect the 2010-2019 dome material.

Although our variably altered samples are from an earlier dome exposed in the summit region, we consider them as representative of the altered dome materials at Merapi volcano (including the newest dome materials). We note, however, that the age of our samples does not relate to their preserved alteration intensity (i.e. we do not consider that the preserved alteration required more than 100 years). We did not collect samples from the active dome because access is not permitted as it is too dangerous, and access to the volcano has been forbidden now for more than a year (the exclusion zone is currently at 3 km from the peak). We allude to why we did not collect samples from the current dome in manuscript:

“In total, five large blocks of lava (M-U, M-SA1, M-SA2, M-HA1, and M-HA2; photographs of the blocks are provided in Figure S1 in the Supplementary Information) were collected in September 2017 from the summit area of Merapi volcano, approximately 100 m to the northeast of the active dome in an area where materials were safely accessible. These blocks, extruded in 1902, were selected as representative of the various degrees of visually discernible alteration present.”

Line 110: Block M-U is the least altered, is highly microfractured, yet according to Figure 1 appears to have some of the lowest permeabilities of all samples studied. So what is this telling us about microfractures influencing permeability?

Since pores are few and far between in sample M-U, these samples would be impermeable without this network of microcracks, which are near ubiquitous in volcanic rocks. The porosity-permeability trend for the (relatively) unaltered rocks in Figure 1 (the grey zone), including the low-porosity, microcracked volcanic rocks with a low permeability (such as samples M-U and M-SA2), has been observed by many other authors, as we state in the manuscript:

“We also note that the permeability of the unaltered dome rock increases as connected porosity is increased (indicated by the grey zone in Figure 1b), in agreement with many published studies for unaltered andesites and basaltic-andesites worldwide (Mueller et al., 2005; Melnik and Sparks, 2002; Bernard et al., 2007; Heap et al., 2014; Farquharson et al., 2015; Heap and Kennedy, 2016).”

The novelty of our manuscript is that the porosity-permeability trends for the highly altered samples are very different (highlighted by the brown and green arrows on Figure 1). Discussing the permeability of the low-porosity, microcracked rocks does not contribute to or alter this narrative, other than to delineate the trend for the unaltered materials. As a result, and because data such as these have been discussed previously, we have chosen not to elaborate on the permeability of block M-U.

Figure 1: The reader cannot easily distinguish among M-U, M-SA2, and M-HA2 because the symbols look so similar. Please fix this.

We agree. We have now changed the symbol colour of M-SA2 to a brighter red (see below). It is now easier to distinguish the three symbols.

Figure 1. We have changed the colour of the symbols of M-SA2 to improve clarity.

Line 198: As you state, there are at least 4 orders of magnitude here, and this is a minimum value. It could be more.

We agree. As stated in the manuscript:

“Thus, we document that acid-sulphate alteration can reduce the permeability of dome rock by at least four orders of magnitude on the sample lengthscale.”

Lines 247-250. Sorry, this makes no sense. You are saying that there is a four order of magnitude change in permeability from 99% sealing to 100% sealing. By 99% sealing the system is essentially closed already, so a 1% increment should not change things.

Although it may at first appear surprising, these numbers are correct. This is because even one high-permeability fracture in a low-permeability rock-mass can have a big impact on the equivalent permeability of a rock-mass. When 99% of the fractures are sealed, there are still four fractures, with a very high permeability of 10^{-10} m², within the modelled rock-mass. The presence of these four fractures results in an equivalent permeability of 10^{-13} m² (thanks to Equation (1)), however, when these fractures are closed, the permeability of the rock-mass drops to the host-rock permeability (which is 10^{-18} m²). We can certainly appreciate that this result is surprising at first, as highlighted by the reviewers query, and it is for this reason why we included the following sentences in the original manuscript:

“Interestingly, reducing the host rock permeability by four orders of magnitude and sealing 99% of the fractures only reduces the equivalent permeability of the dome by about two orders of magnitude. When 100% of the fractures are sealed, however, the permeability of the dome is reduced to 9.2×10^{-18} m², highlighting the importance of few, or even isolated, fractures in maintaining the high dome permeability required for efficient outgassing of the underlying magma-filled conduit.”

Line 258: You do not justify your choices of conduit and edifice permeability, which have a huge difference. I could easily argue that the values should be reversed. The edifice should be highly permeable, especially at Merapi which is built of pyroclastic material. The conduit is a plug of magma which may have extremely low permeability.

We agree with the reviewer that the assumed values of permeability for the magma-filled conduit and edifice could be different. We hope that the reviewer agrees that it is a nontrivial task to decide on values of permeability for such edifice-scale modelling, and that there will be of course arguments for choosing values that are either higher or lower than those chosen. Volcanic systems are, of course, extremely complex and modelling always requires some degree of simplification. We reassure the reviewer that our input parameters were not chosen lightly, and were based on the results of many simulations and several discussions between the co-authors of the manuscript. Importantly, and the most challenging aspect of our modelling, is that, to assess the influence of solely changing the permeability of the dome, we were required to fix the values of permeability for the magma-filled conduit and edifice (although ancillary simulations are presented in the Supplementary Materials). The justification for our choices is discussed below.

First, we would expect the edifice (of Merapi volcano and other andesitic stratovolcanoes) to be constructed from the products of successive explosive and effusive eruptions (rather than just permeable pyroclastic material) and, since the permeability of layered strata can be considered as a circuit in series (i.e. the fluid must pass through each layer to escape), the equivalent permeability of a system is often closest to that of the unit with the lowest permeability. Since the permeability of volcanic rocks can be highly variable, we chose what we considered as an “intermediate” and “reasonable” value for the edifice rock, based on the experimental data presented in, for example, Farquharson et al. (2015). We also have to consider that laboratory permeability values for volcanic rocks will also underestimate the permeability of a volcanic rock-mass,

which invariably contains networks of permeability-enhancing fractures. With this in mind, we consider 10^{-13} m^2 as a reasonable estimate of the permeability of a volcanic edifice (i.e. the host-rock). Further, and as mentioned above, ancillary simulations, in which we vary the edifice permeability, are presented in the Supplementary Materials.

Second, although our choice of permeability for the magma-filled conduit may not represent the permeability of a conduit plug which, as the reviewer points out, will likely be characterised by a lower permeability, we consider our choice as a “reasonable” value for a conduit containing a batch of fresh volatile-rich, bubbly magma. We also highlight that our chosen permeability value does not represent the permeability of the magma, but the equivalent permeability of the magma-filled conduit. This value therefore considers the idea, for example, that the conduit is likely enveloped by a brecciated or fractured zone of high permeability.

To avoid any future confusion, we consider it important to clearly state that our chosen permeability values represent the “equivalent permeability” of these zones. For example, we consider it more appropriate to use the term “the equivalent permeability of the magma-filled conduit”, rather than “the permeability of the magma”. We have now reworded the following sentence in the manuscript to improve clarity:

“The model was split into three domains: the **magma-filled** conduit, the edifice, and the lava dome (see Figure 3a). To assess the role of dome permeability, we fixed the **equivalent permeability of the magma-filled** conduit and edifice at, **respectively, 10^{-10} and 10^{-13} m^2** , and varied the **equivalent permeability of the** lava dome from 10^{-11} to 10^{-13} m^2 ...”

Figure 3: The figure is labelled “host rock” but what you mean here is “edifice” do you not? Otherwise the reader is confused by “host rock” referring to dome material on lines 239-243, and host rock” referring to the edifice in Figure 3.

The reviewer is correct. We’ve now changed “host rock” on the figure to “edifice” (see below). We have also changed “conduit” to “magma-filled conduit”.

Figure 3. We have now changed the label “host rock” to “edifice” on panel (a).

Lines 315-330: The best monitoring evidence for sealing are changes in SO₂ flux as a function of time. Merapi is a well monitored volcano for which SO₂ data are certainly available.

We thank the reviewer for this comment. We’ve given it serious consideration and the co-authors have discussed at length the availability of such data. What we have found is as follows:

- 1. COSPEC (correlation spectrometer) data are available from 1992-2011 in a 2017 report (written in Indonesian). However, measurements were typically only taken a couple of times a month and there are several long periods (many months) for which no data were collected. Although these data are useful for understanding long-term trends in SO₂ output at Merapi volcano, we do not consider them of sufficient resolution to test our hypothesis (which would require high resolution SO₂ data prior to an explosion/eruption).**
- 2. From 2010, a permanent DOAS (differential optical absorption spectrometer) network was installed at Merapi volcano, which takes high-resolution SO₂ data. However, although some data are shown in reports, there are no peer-reviewed papers containing these data (that we are aware of) and the details on the data processing and filtered are not explained in the available reports.**

Due to the problems described above, we conclude that, unfortunately, and despite the various monitoring initiatives, we are unable to test our hypothesis using SO₂ data

collected at Merapi. However, in the absence of peer-reviewed data from Merapi, we can discuss published SO₂ data from other volcanoes worldwide. In fact, there are several studies (e.g., Stix et al., 1993; Edmonds et al., 2003; Campion et al., 2018; de Moor et al., 2019) that show a decrease in pre-eruptive SO₂ flux, which is consistent with our proposed mechanism. We have now included the following sentence:

“Although high temporal resolution SO₂ flux data are currently unpublished for Merapi volcano, a reduction in pre-eruptive SO₂ flux has been observed at, for example, Galeras volcano (Stix et al., 1993), Soufrière Hills volcano (Montserrat) (Edmonds et al., 2003), Popocatépetl volcano (Mexico) (Campion et al., 2018), and Poás volcano (de Moor et al., 2019), lending support to the mechanism outlined in Figure 4.”

We again thank the reviewer for prompting this addition, which we consider has further strengthened the argument for the mechanism proposed in our manuscript.

Line 360: Replace “and/or” with “and”.

Changed as suggested.

Reviewer #2

The authors did a nice job of addressing the comments in the 1st round of reviews. The revised study is now more convincing. This is a thought-provoking manuscript that I am happy to endorse for publication.

We're pleased that Reviewer #2 considers our work "thought-provoking" and now endorses our manuscript for publication.

Reviewer #3 (Cristian Montanaro)

Dear Dr. Heap and colleagues,

The submitted and revised manuscript "Hydrothermal alteration of andesitic lava domes leads to explosive volcanic behaviour" include all changes in response to my comments. Regards the rebuttals, the authors decided not to modified some of the points I was making, but their justification/argumentation were valid and well argued, thus I am fine with the authors responses. Some of the manuscript paragraphs/section I was commenting, now appear further improved also thanks to the comments of the other referees. Overall, I do not have further comments/corrections on the submitted version which, in my opinion, is to be totally accepted for publication.

We're pleased that Reviewer #3 considers that our manuscript should now be "totally accepted for publication".

Only a minor thing, please change my name in the acknowledgments to Cristian without "h".

We thank the reviewer for spotting this typo, which has now been corrected.

REVIEWERS' COMMENTS:

Reviewer #4 (Remarks to the Author):

General Remarks

Overall, I find this paper very well written, easily understandable, and well-reasoned. The authors have thoroughly addressed all the comments provided by Reviewer #1, and I find their changes have improved the clarity of the paper significantly. I recommend this paper for publication with very minor final edits for clarity.

On a personal note, this paper directly validates the discussion points made in Ball et al. (2013), a paper resulting from my PhD studies. I am pleased to see that colleagues have taken the next steps in experimental and modeling research to confirm the possibility that alteration-based sealing could contribute to lava dome overpressure and instability.

Detailed comments:

Line 28: Add 'distances of' or similar phrase to clarify that this sentence is referring to the scale of the dome rock.

Line 31: Would be nice to include in the abstract the actual factor of increase of the pore pressure (or at least a qualitative description).

Line 37: Add 'continuous' to 'gas monitoring' to emphasize that sporadic SO₂ measurements aren't sufficient, as stated later in the manuscript.

Line 59: This would be an excellent place to also cite Horwell et al. (2013) (<https://doi.org/10.1007/s00445-013-0696-3>), who concluded that cristobalite mineralization in lava domes could reduce porosity, permeability, and therefore dome stability. Cristobalite is not part of the range of minerals associated with acid-sulfate alteration, but is still a secondary phase that plays the same role in reducing permeability.

Line 69: This paragraph is one long sentence – please break it up for clarity.

Line 245: Replace 'host' with 'edifice' as per the response to Reviewer #1's comments on Figure 3.

Line 259. Remove 'augmentation' (unnecessary).

Figure 1. Missing the grayscale colors in the markers for unaltered samples in 1c. They are present in 1a.

Jessica L. Ball

Reviewer #4 (Jessica Ball)

General Remarks

Overall, I find this paper very well written, easily understandable, and well-reasoned. The authors have thoroughly addressed all the comments provided by Reviewer #1, and I find their changes have improved the clarity of the paper significantly. I recommend this paper for publication with very minor final edits for clarity.

We're pleased by these endorsements.

On a personal note, this paper directly validates the discussion points made in Ball et al. (2013), a paper resulting from my PhD studies. I am pleased to see that colleagues have taken the next steps in experimental and modeling research to confirm the possibility that alteration-based sealing could contribute to lava dome overpressure and instability.

And it is for this reason why we are pleased that Dr. Jessica Ball has reviewed our manuscript. The authors consider her papers (from 2013 and 2015, both cited in the submitted manuscript) as important works in first highlighting the potential influence of hydrothermal alteration on the permeability and stability of a lava dome. As Reviewer #4 states, our manuscript uses new data and modelling to explore these testable hypotheses and we show quantitatively that hydrothermal can indeed promote explosive volcanic behaviour.

Detailed comments:

Line 28: Add 'distances of' or similar phrase to clarify that this sentence is referring to the scale of the dome rock.

Based on the strict word limit (< 150 words) for the abstract, these parentheses have now been removed.

Line 31: Would be nice to include in the abstract the actual factor of increase of the pore pressure (or at least a qualitative description).

Based on the strict word limit (< 150 words) for the abstract, we cannot elaborate on the increase in pore pressure. But, we do state in the following sentence that this increase is sufficient to fragment the majority of dome-forming materials.

Line 37: Add 'continuous' to 'gas monitoring' to emphasize that sporadic SO₂ measurements aren't sufficient, as stated later in the manuscript.

Based on the strict word limit (< 150 words) for the abstract, we deleted the mention of muon tomography, electrical tomography, and gas monitoring. We now simply state:

"It is crucial that hydrothermal alteration, which develops over months to years, is monitored at dome-forming volcanoes and is incorporated into real-time hazard assessments."

Line 59: This would be an excellent place to also cite Horwell et al. (2013)

(<https://doi.org/10.1007/s00445-013-0696-3>), who concluded that cristobalite mineralization in lava domes could reduce porosity, permeability, and therefore dome stability. Cristobalite is not part of the range of minerals associated with acid-sulfate alteration, but is still a secondary phase that plays the same role in reducing permeability.

We thank the reviewer for this suggestion. We now cite Horwell et al. (2013).

Line 69: This paragraph is one long sentence – please break it up for clarity.

This has now been split into two sentences:

“It is understood that volcanic character, effusive versus explosive, depends on many interconnected parameters⁸. Magma flow rate, for example, will dictate the time available for outgassing, cooling, and crystallisation that, in turn, influence magma viscosity⁸ and the resultant dome morphology, including the number density and morphology of fractures within the dome²⁷. The goal of this contribution is to quantitatively assess whether hydrothermal alteration alone is sufficient to promote explosive volcanic behaviour.”

Line 245: Replace ‘host’ with ‘edifice’ as per the response to Reviewer #1’s comments on Figure 3.

Here, “host rock” is correct. The “host rock permeability” is the permeability of the intact “host” rock (i.e., the intact rock between the fractures) whereas the “edifice permeability” refers to the equivalent permeability of the edifice (host rock plus fractures).

Line 259. Remove ‘augmentation’ (unnecessary).

The word “augmentation” has now been removed.

Figure 1. Missing the grayscale colors in the markers for unaltered samples in 1c. They are present in 1a.

In fact, the symbols for the unaltered samples in Figure 1c should all be white (as these samples are all from Kushnir et al., 2016). The unaltered samples are split into three groupings (indicated in the legend in Figure 1b): “2006” (this study, grey), “M-U” (this study, black), and “Kushnir et al. (2016)” (from Kushnir et al. (2006), white).

Jessica L. Ball